# Patent Landscape of Composting Technology: A Review

**Fatin Amanina Azis** *, **Masrur Rijal**, **Hazwani Suhaimi** and **Pg Emeroylariffion Abas** *

Faculty of Integrated Technologies, Universiti Brunei Darussalam, Gadong BE1410, Brunei;
18b3103@ubd.edu.bn (M.R.); hazwani.suhaimi@ubd.edu.bn (H.S.)
* Correspondence: 20h8455@ubd.edu.bn (F.A.A.); emeroylariffion.abas@ubd.edu.bn (P.E.A.)

**Abstract:** Organic waste management is a major global challenge. It accounts for a significant portion of waste that ends up in landfills, where it gradually decomposes and emits methane, a harmful greenhouse gas. Composting is an effective method for potentially solving the problem by converting organic waste into valuable compost. Despite many studies focusing on the composting process, no study has reviewed the technological advancements in the composting fields from the perspective of patents. This review paper begins with background information on the composting process, specifically important factors affecting the process, problems associated with it, and the available technologies to facilitate the process. Different technologies are discussed, ranging from manual to automated methods. Subsequently, 457 patents are selected, classified into different categories, and reviewed in detail, providing a patent technology landscape of composting technology. Automatic composters are more prominent than manual ones as managing organic waste at the source has become more crucial in recent years. The need for a domestic composter creates an opportunity for the development of a compact and automated system for organic waste management, which is more suitable for urbanized settings. This technology has the potential to reduce the amount of organic waste that needs to be managed at an already overburdened landfill, as well as the environmental consequences associated with it.

**Keywords:** organic waste; electric composter; composting technology; automatic; waste management

## 1. Introduction

Waste management is a major global issue that, if not appropriately addressed, could present significant future global consequences. The rapid growth of the world's population, combined with increased urban development in search of improved living standards, has resulted in a daily increase in the production of waste that shows no signs of abating. According to estimates, global municipal solid waste generation will increase by approximately 70% by 2050, reaching 3.4 billion metric tons [1]. Naturally, governments are becoming increasingly concerned about the increased production and accumulation of waste, as well as its potential impact on the environment, and are thus exploring various methods of mitigating the problem. Several waste management methods, including landfilling, incineration, and recycling, have been implemented [2]; the most common being the burial of waste, particularly municipal solid waste, in landfills.

Landfills have risen to become one of the most widely used waste disposal methods in recent years. Due to its low cost and low technical requirements, open dumps or landfills are commonly used to dispose of waste in approximately three-quarters of countries around the world. However, the complete decomposition of waste in a landfill may take several years [3], resulting in an accumulation of enormous piles of waste and putting a strain on the already limited disposal cells at the landfill. The piling of over-accumulated waste in overfilled landfills creates an oxygen-deprived environment, causing anaerobic decomposition to take place. Biogas, a mixture of carbon dioxide and methane, is commonly produced and released in large quantities into the atmosphere during the anaerobic decomposition process, and this is especially true if the waste is primarily composed of

organic waste. Methane gas is 28 times more potent than carbon dioxide [4]. Consequently, these generated biogases contribute significantly to climate change and form a big part of the problem of global warming [5]. According to a report by the International Solid Waste Association (ISWA), landfills are expected to generate 10% of greenhouse gas emissions by 2025, and if no remediation is done to control the release of biogas from landfills, this figure will surely rise even higher [6].

Solid waste incineration is commonly used as a quick way of reducing the rapidly growing amount of waste while also generating energy, particularly in large cities. However, incineration is costly, inefficient, and harmful to the environment [5]. Dangerous air pollutants, including fine dust, carbon monoxide, acid gases, nitrogen oxides, and carcinogenic dioxins, as well as hazardous waste associated with fly and bottom ashes, are produced by the burning of wastes in incineration plants [7]. The production of hazardous waste necessitates careful handling and disposal. Furthermore, incineration may also cause water pollution, odors, noise, and vibrations, which can have significant impacts on nearby residential and business areas.

Another approach in managing solid waste is through recycling [8]. Different types of waste may require different recycling methods; with solid waste commonly categorized into plastic, metal, paper, glass, organic waste, and others [9], before converting them back into raw materials that can be reused to make new and valuable products [10]. Consequently, this reduces the demand for new raw materials to produce glass, paper, metal, and plastic products. In many cases, the manufacture of goods from recycled materials may require less energy than manufacturing the goods from raw materials. Furthermore, recycling also keeps materials out of the landfill in the first place, and hence, reduces greenhouse gas emissions associated with landfills. In fact, recycling is one of the safest and most effective methods of managing waste.

Figure 1 depicts global waste composition, showing that waste is mainly composed of food and green waste, referred to as organic waste, accounting for 44% [11] of the overall waste composition. Several methods may be used to recycle organic wastes, thereby preventing them from being sent to landfills, including using them as a source of animal feed and through composting processes. Some organic wastes, particularly green wastes, may be used as feeds for farm animals; however, using them as feeds may be more applicable to farmers and animal breeders only. Composting is the most preferred method for managing organic waste, as it applies to the masses, does not require significant areas, and of course, is capable of reducing the rate of the production of waste, while at the same time, producing valuable by-products, in the form of compost.

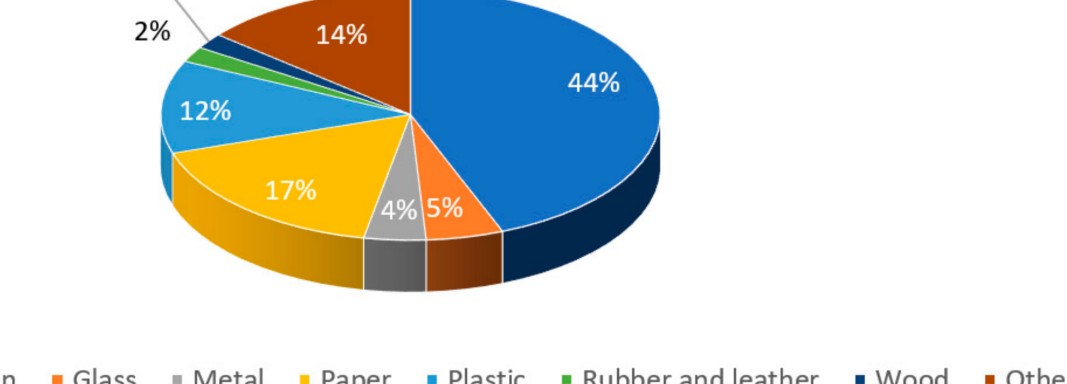

**Figure 1.** Percentage of global waste composition.

There are generally two main types of composting: aerobic and anaerobic composting. In aerobic composting, the decomposition of organic waste occurs in the presence of oxygen, whilst oxygen is absent during the decomposition process in anaerobic composting. Anaerobic digestion is commonly done in a specialized container, called a digester. Even though the process can produce fertilizers that can be used in agriculture, and biogases [12], which can be used to generate electricity and heat, or processed into renewable natural gas, the method requires a significant investment and a high labor force. A digester is an expensive air-tight vessel designed to retain gas and requires regular maintenance for optimized operation and safety. As such, aerobic composting may be the best alternative for managing organic waste.

However, composting is known to be a time-consuming process that can take months or even years to fully complete the decomposition process and produce compost. Despite producing carbon dioxide as a by-product, the amount produced is relatively smaller than the production of more harmful gases by other modes of waste management, and hence, it can be considered environmentally friendly. An average compost pile measures approximately $3 \times 3$ feet and necessitates regular manual turning for aeration and temperature-control purposes. To facilitate the process, an automated electrical composter has been suggested, commonly designed with a grinder to reduce waste size, automatic turning, as well as multiple sensors, to ensure optimum composting conditions. An average electric composter takes five hours to decompose food waste, although some electric composters may only require 48 h or a few days, which is much faster than conventional composting. An automatic composter is also commonly designed to be compact, making it suitable for domestic management of organic waste at the source. Ultimately, in terms of environmental impact, an electrical composter emits less gas than other organic waste management methods, pointing to its value in the waste management chain. Although many studies have been carried out to understand the concept of composting processes, none have reviewed patents relating to technological advancements in composting. Thus, there is a clear need for a patent review of electrical composting to illustrate the technological advancements in the field from the patent perspective.

## 2. General Processes of Composting

### 2.1. Anaerobic Composting

In anaerobic composting, decomposition of the organic waste occurs in the absence of oxygen, whereby anaerobic microorganisms or anaerobes dominate the process to produce biogases, mainly methane gas, and slurry compound with a strong odor called digestate, as by-products [13]. Naturally, the digestate by-products accumulate over time, as they are not further metabolized. Since anaerobic composting is a process occurring at low temperatures, weed seeds and pathogens remain intact, with the process commonly taking a longer time than aerobic composting. Figure 2a shows the basic block diagram of anaerobic composting; the processing of organic waste with the help of anaerobe microorganisms in the presence of moisture, and producing digestate and biogas in the process. Anaerobic composting can be presented by Equation (1), [14],

$$C_aH_bO_cN_d + \left(\frac{4a - b - 2c + 3d}{4}\right)H_2O \rightarrow \left(\frac{4a + b - 2c - 3d}{8}\right)CH_4 + \left(\frac{4a - b + 2c + 3d}{8}\right)CO_2 + dNH_3 \qquad (1)$$

Biogas output from the anaerobic digestion may also be utilized, using a digester tank that can store the gas to be used as fuels for generating electricity or heat. However, the use of a digester tank for storage and the use of methane, require high investment costs, high maintenance costs, and are labor-intensive [15].

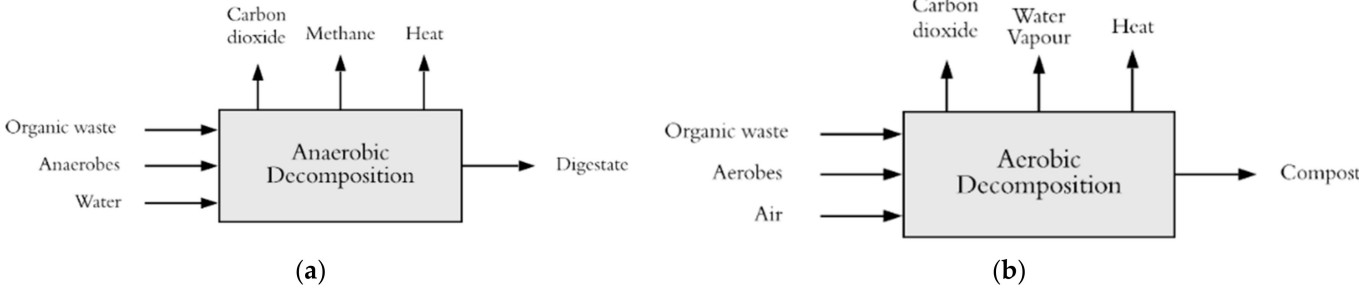

**Figure 2.** (**a**) Basic block diagram of anaerobic composting; (**b**) basic block diagram of aerobic composting.

### 2.2. Aerobic Composting

On the other hand, aerobic composting is a controlled decomposition of organic waste in a sufficient amount of oxygen to convert the organic wastes into nutrient-rich composts [16]. Several factors need to be properly controlled to effectively create and manage a good compost heap, and hence, produce desirable composts. These factors include the population of microorganisms in the compost heap, as well as its conditions, including temperature, moisture, oxygen, and soil pH [17], which commonly have ideal ranges for optimum aerobic composting to occur. Other important factors also include carbon to nitrogen ratios of the organic matters [18], as well as porosity [19]. The aerobic organisms, including fungi, microbes, actinomycetes, and invertebrates [20], are relied upon for the composting process to effectively break down the organic wastes. These microorganisms consume and utilize organic compounds in the organic wastes for their well-being; particularly, carbon for energy, nitrogen for protein building, and other compounds, including phosphorus and sulphur, for other cellular activities, to produce the desired humic-like composts, whilst at the same time producing carbon dioxide, heat, and water vapor as by-products during the process [21]. Aerobic composting is shown in Figure 2b, with the mathematical representation given by Equation (2), [14],

$$C_aH_bO_cN_d + \left(\frac{4a + b - 2c + 3d}{4}\right)O_2 \rightarrow aCO_2 + \left(\frac{b - 3d}{2}\right)H_2O + dNH_3 \quad (2)$$

Generally, aerobic composting is a much faster process than anaerobic composting [22] and has fewer environmental impacts when adequately controlled. The heat generated during aerobic composting is usually adequate to kill harmful bacteria and pathogens [23], producing valuable compost. These make aerobic composting the most efficient and environmentally friendly method of managing organic waste.

Aerobic composting can be characterized by its three distinct phases: the mesophilic, thermophilic, and maturation phases [24]. During the mesophilic phase, which occurs at a moderate temperature, high consumption of amino acids from the organic wastes by the microorganisms leads to the rapid growth of mesophilic microorganisms. In turn, this causes the temperature of the pile to significantly increase, up to the point of destroying the mesophilic microorganisms themselves, only to be replaced by thermophilic microorganisms. During the thermophilic phase [25], thermophilic microorganisms dominate the pile, with only a few mesophilic organisms surviving the high temperature. Collectively, the mesophilic and thermophilic phases are referred to as the active stage [19], which can last several weeks. The majority of composting occurs during this active stage, with plant wall elements, including cellulose and hemicellulose, broken down into humic material and carbon dioxide [26]. The compost heap is sanitized once the temperature reaches 70 °C [27], as naturally occurring pathogens inside the compost heap are eliminated by the high temperature. Clearly, temperature plays an important role and needs to be controlled for the mesophiles and thermophiles to conduct their work optimally. Moisture is also necessary during the composting process; as materials may decompose slowly if a pile is too dry, necessitating the addition of a considerable amount of water [28], while a too damp pile needs to be rotated and mixed, to reduce the moisture content. The compost

heaps also require regular turnings depending on the size of the pile to bring sufficient oxygen to the inner pile to facilitate the breakdown of material by the bacteria and for odor control [25]. Smaller particles also help maintain optimum temperatures by producing a more homogenous compost mixture and improving pile insulation [29], while fine-sized particles may hinder air from freely passing through the pile and thereby slowing down the decomposition process.

As resources become depleted due to the decomposition by the microorganisms, the process begins to slow; resulting in temperature drops [30]. This is referred to as the curing stage, which is generally longer than the active stage and can last several months. Mesophilic microbes are once again thriving, displacing the majority of thermophiles, as the compost heap begins to cool and mature [20], eventually becoming a humic substance referred to as compost that can be used as a source of nutrients for other soil applications.

Composting can be a timely process, with the composition of waste, the availability of beneficial microorganisms, and composting environment determining its success. The size of the compost heap, the types of organic materials, the surface area of the materials, and the number of times the compost is turned, may influence the amount of time it takes to produce a good compost [31]. For the compost heap to have a good range of temperature for heat retention, the size of the open space has to be at least one cubic yard [32]. A good mixture of dark organic material, such as dry leaves, twigs, and manure, and green organic material, such as grass clippings, need to be combined to give a good compost. Generally, brown compounds are carbon-rich, while green materials are nitrogen-rich. A big compost heap needs to be turned more regularly to supply ample oxygen to the center of the heap [16], as well as to limit too high of a temperature, which may kill the beneficial microorganisms. This is in contrast to smaller compost heaps, which do not require regular turnings, as oxygen can naturally reach the center of the heap. Moreover, the surface area of the organic waste may be increased to provide more surface area for the microorganisms to feed by grinding, chipping, and shredding the organic materials [33].

However, anaerobic conditions can also occur if the environment is not adequately controlled during an aerobic composting process. Primarily, lack of oxygen may cause the anaerobic microorganisms to thrive, leading to the compost heap to turn anaerobic, with the emissions of greenhouse gases, particularly methane gas, into the air, which can harm the environment. As such, it is important to properly control the aerobic composting process by providing a conducive environment for the aerobic microorganisms to thrive.

## 3. Important Parameters in the Composting Process

The presence of beneficial microorganisms, nutrients, and conducive conditions for microbial activities, are the basic components of an aerobic composting process [34]. Aerobic composting requires an abundance of both mesophilic and thermophilic microorganisms at different composting stages. These microorganisms require nutrients for energy and protein-building, which are provided by the compost heaps, and these nutrients need to be provided at optimal conditions in terms of porosity, availability of oxygen, temperature, moisture, and soil pH [35]. In essence, aerobic composting is an exercise in the breeding of beneficial microorganisms by providing them with sufficient nutrients and optimal conditions, such that they will be able to produce a good and mature compost over time. This section discusses the essential factors that need to be considered in an aerobic composting process.

### 3.1. Carbon to Nitrogen (C:N) Ratio

Composting necessitates a proper balance of feedstock or input of green and brown organic components [36]. Green organic materials, including grass clippings, food scraps, and manure, contain a lot of nitrogen and are used in the biosynthesis of proteins, enzymes, and nucleic acids, for the growth and cell functions of the microorganisms. On the other hand, brown organic materials, including dry leaves, wood chips, and branches, contain a lot of carbon. These are used as a source of energy by the microbes during the composting process. Generally, carbon is consumed much quicker than nitrogen, and

consequently, higher carbon content (brown materials) is required in proportion to nitrogen (green materials).

However, too much carbon with insufficient nitrogen in the compost heap may inhibit cell growth and the overall microbial development process, and it may even cause some microbes to die [16]. To complete the nitrogen cycle and continue the decomposition process, the microbial cells may pull the available nitrogen from the soil in proportion to the available carbon for decomposition, known as soil robbing. On the other hand, in the event of an insufficient amount of carbon in the compost heap, surplus nitrogen would be excreted in the form of ammonia ($NH_3$) once the carbon has been exhausted [37]. Ammonia gives out a distinctive strong urine-like odor. These losses of nitrogen from the soil and compost heap cause insufficient nitrogen and carbon in the compost heaps, resulting in a reduction of nutrients in the compost, and hence, should be avoided to the greatest extent possible. As such, there is a need to strike a balance and only put the right amount of carbon and nitrogen in the compost heaps. The literature has highlighted that a carbon to nitrogen mass ratio (C:N mass ratio) of between 25:1 and 30:1 is the optimum range [38].

*3.2. Particle Size*

Generally, decomposition by the microorganisms during the composting process occurs superficially on the surface of the compost heaps' particles. Smaller compost particles increase the surface area for the decomposition and may speed up the decomposition process. Microorganisms in aerobic composting require oxygen to decompose the organic waste [39], with the volume of air inside a compost heap referred to as its air-filled porosity. Very small compost particles typically result in the compacting of the compost heaps, resulting in reduced oxygen and moisture. Again, balance in particle size needs to be right. Small particle size provides a large surface area for decomposition, but may deprive the microorganisms of the much-needed oxygen due to the low air-filled porosity. On the other hand, large particle size provides plentiful oxygen, but limits the surface area for the decomposition process. Ideally, porosity should be maintained between 35 and 50 percent of the volume of the compost material [40,41] to allow air to pass through the structurally porous compost heap and cross the water layer barrier, providing air to the bacteria residing on the particle's surface.

Commonly, air-filled porosity is usually inferred from its bulk density and moisture content rather than tested directly. To obtain an optimum air-filled porosity, it is common to have a mixture of smaller and larger-sized particles. Additionally, adding bulk agents, including wood chips and shredded newspapers, is common to enhance porosity. Figure 3 below illustrates the porosity structure of compost particles.

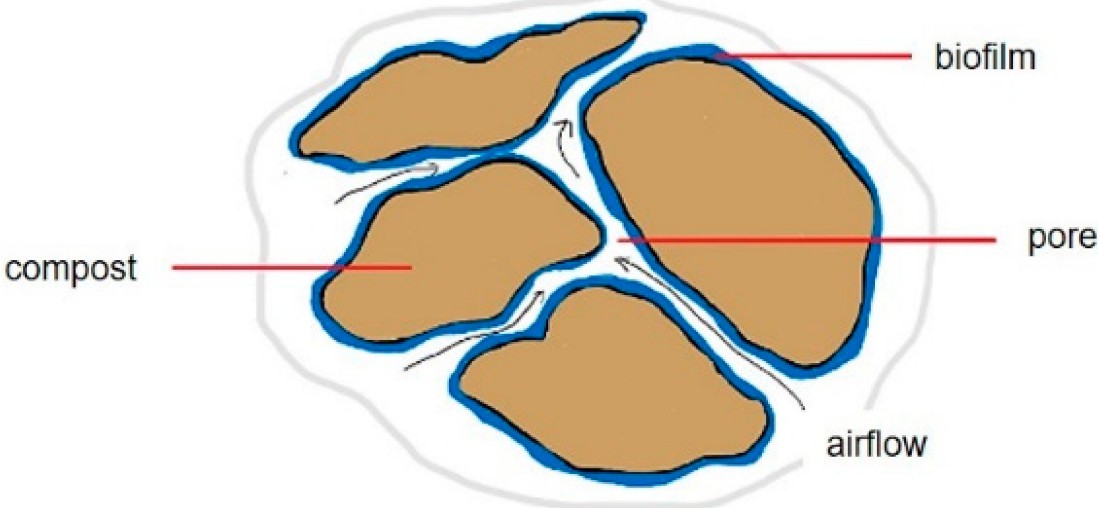

**Figure 3.** Porosity of compost.

### 3.3. Temperature

Microorganisms require a specific temperature range to function properly and optimally [42]. As previously described, anaerobic composting can be characterized by the different thermal phases, with different microorganisms dominating each phase. The mesophilic microorganisms, which thrive below 40 °C, dominate both the mesophilic and maturation phases, while the thermophilic microorganisms, which thrive between 40 and 70 °C, dominate the thermophilic phase. The temperature of the pile's core can rise up to 70 °C due to the microbial activity [19], with the high temperature capable of killing most pathogens and hence, sanitizing the compost heap. However, too high of a temperature may also kill the beneficial thermophilic microorganisms, which should be avoided.

The biological activity of the composting heap is heavily influenced by its temperature. Balance of greens, browns, air, and water create optimum conditions for aerobic microorganisms to develop and prosper, resulting in hot composting. Physical and chemical interactions occurring within the pile allow heat to be generated internally during the natural decomposition process, with an optimum temperature of greater than 40 °C during the thermophilic phase for sanitation purposes [43]. However, un-balance in one or more of the components necessary in an aerobic composting process may develop anaerobic conditions, with the temperature not rising sufficiently at the different phases of composting.

### 3.4. Oxygen Concentration and Moisture Content

Oxygen and moisture are two important components in the composting process. Oxygen concentration in a compost heap practically determines whether the reaction will be an aerobic or anaerobic process. Particle size and air-filled porosity play significant roles in determining the amount of oxygen that can reach the microorganisms during the decomposition of organic wastes, with a minimum of 5% concentration of oxygen providing the ideal amount of oxygen in a compost heap [19,44]. Microorganisms in a compost heap also require an optimum moisture level for them to flourish, with the theoretical ideal moisture content of around 45 to 60% by dry weight [19,45]. The moisture acts as a transport medium and is necessary to allow the microorganisms to access organic matter nutrients. Just enough moisture should only be provided to hydrate a compost heap, as over-watering may suffocate the microorganisms of air, resulting in anaerobic conditions to develop, which slows the decomposition process, causes foul odors, and results in the release of harmful methane gas [39].

### 3.5. pH Value

Compost microorganisms thrive in neutral to acidic environments, with pH levels ranging from 5.5 to 8 during the composting process. However, pH values of compost heaps at different points within the heap may be inhomogeneous and likely to change, so several replications of measurements need to be considered. Organic acids are produced during the mesophilic phase, and as fungi prosper in acidic environments, the slightly acidic pH aids in the breakdown of lignin and cellulose [46]. However, organic acids are neutralized as composting progresses into the thermophilic phase, with a typical pH value of six to eight during the maturation phase [19]. However, organic acids may build up rather than be broken down during composting if anaerobic conditions occur. Acidity may generally be reduced by aerating or mixing the system.

### 3.6. Summary of the Optimum Values for the Composting Parameters

Table 1 summarizes the important parameters in a composting process and their corresponding ideal range of values. A combination of beneficial microorganisms, organic materials, and optimum environmental conditions is vital for producing good compost within a suitable time frame. A balanced approach must be taken, be it in providing the material compositions or in providing the optimum environment for the composting process to be smooth and efficient.

**Table 1.** Ideal range of several important factors of composting.

| Variable | Ideal Range |
|---|---|
| Moisture | 45–60% by weight |
| Oxygen concentration | >5% |
| Temperature | 40–65 °C for thermophilic phase |
| C:N ratio | 25:1–30:1 |
| Porosity | 35–50% |
| pH | 5.5–8.0 |

## 4. Different Composting Technologies

Aerobic composting can be performed on varying scales [47]. Small-scale composting is commonly done domestically, using organic wastes produced in the kitchen and channeling the resulting compost to personal gardens and backyards. This represents a low-cost, effective, and easy method of handling a small amount of daily kitchen waste; overall, reducing the amount of household waste that needs to be sent for further processing. On the other end of the scale, large-scale aerobic composting may require a large space and resources; and a specialized plant or factory may be needed for the operation. This requires a significant investment and necessitates proper collection of often mixed wastes, effective separation and segregation of the organic wastes from the source, and other pre-processing before the processed wastes can be fed into the specialized aerobic composting plant. It is evident that small-scale domestic aerobic composting can be a part of the overall waste management solutions; by contributing to an overall reduction of organic waste going into landfills, whilst producing by-products, which are beneficial to individuals as well as the agricultural sectors [48].

Composting can be performed manually or automatically. Manual composting technology is primarily a natural process and commonly requires a relatively long time to fully decompose to produce the final compost product [16]. Additionally, it requires the availability of outdoor spaces that are relatively isolated to prevent odor, and has to be monitored regularly to ensure optimum composting conditions. On the other hand, an automatic composter attempts to automate some of the phases of the natural composting process. Generally, it requires less monitoring and occupies less space, making it more suitable for the urbanized setting. However, it commonly requires electricity to speed up the composting process. The following sections explore some of the manual and automatic composting technologies found in the literature.

### 4.1. Manual Technology

In manual composting, the process is operated by hands and through mechanical means without automation. Five common types of manual composting methods: windrow, passively aerated windrow, bin, in-vessel, and vermicomposting methods, are described.

### 4.1.1. Windrow Composting

Windrow is the general term for the use of an elongated pile of stacked raw organic materials for composting, as shown in Figure 4a, and represents the most basic composting method. The method is suitable for treating large volumes of organic waste and producing large volumes of compost [49]. However, due to its simplicity, it is one of the most commonly adopted manual composting methods, especially for domestic composting.

Due to the sometimes-large windrow compost heaps, air-filled porosity needs to be maintained throughout the compost heaps, and aeration becomes a very important issue [50]. Commonly, the compost heaps are mixed with structure-giving materials, such as twigs, cardboard, or hard vegetables, to allow air to pass throughout the compost heaps effectively and oxygen, which is consumed by the microorganisms, to be replenished [51]. Particles in the compost heaps may also be reduced, either manually or automatically using a grinding machine, to increase surface area and hence, allow faster decomposition. In any case, the compost heaps need to be manually or mechanically turned [52] to re-

establish porosity over time and re-introduce air and oxygen back into the compost heap. Additionally, turning also allows the rotation of compost materials, such that the exterior of the heap can be rotated to the interior of the heap and allow the microorganisms to inter-changeably decompose different parts of the heap.

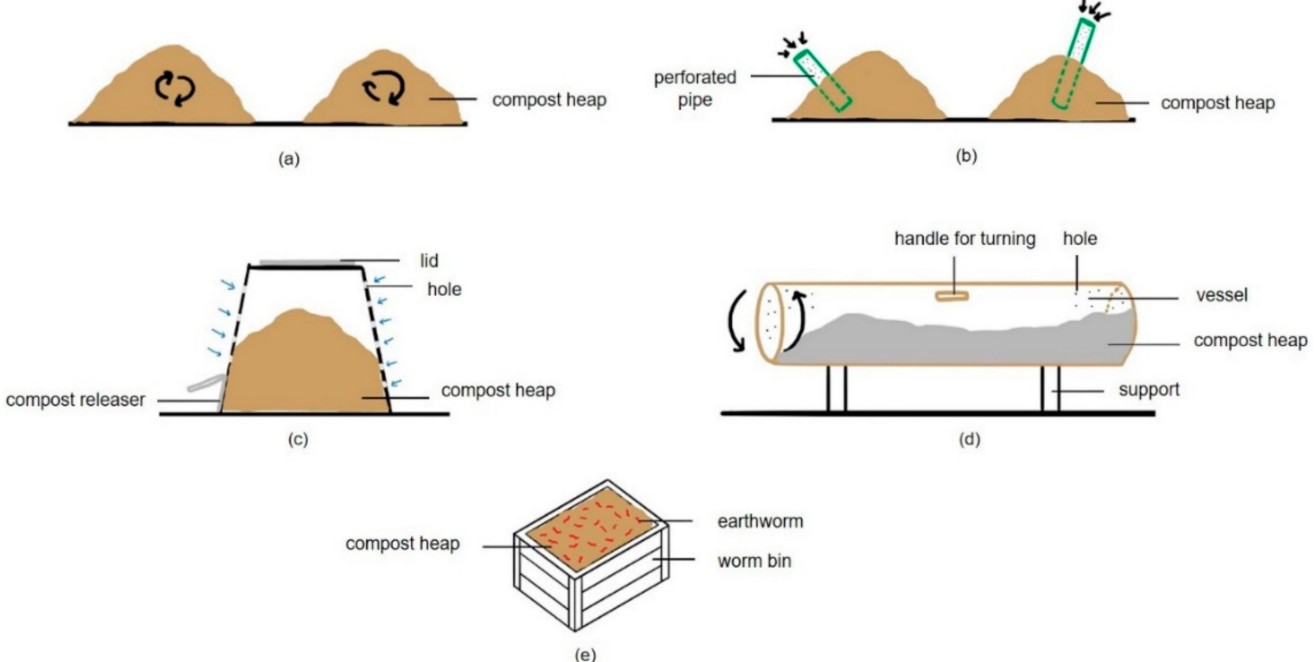

**Figure 4.** (**a**) Windrow composting; (**b**) passively aerated windrow composting; (**c**) bin composting; (**d**) in-vessel composting; (**e**) vermicomposting.

The obvious benefits of windrow composting are that it requires low funding and maintenance [53]. However, it needs an amount of space to accommodate the compost heaps, which need to be spaced out from one another, for an effective composting process. Composting using the windrow method also consumes a relatively longer amount of time to produce good compost and requires manual labor for reducing the particle size to the appropriate size and turning.

### 4.1.2. Passively Aerated Windrow Composting

As shown in Figure 4b, passively aerated windrow composting is an improvement over simple windrow composting by introducing perforated pipes to allow convection airflow throughout the organic compost heaps, particularly at the center of the heaps [54]. Different pipe configurations can be adopted to bring oxygen to the microorganisms, with the use of the pipe eliminating the need for frequent turning. However, it is important to introduce the right air-filled porosity before putting the organic waste in the compost heap by appropriately reducing the compost particles as well as thorough pre-mixing. Insulation of the compost heap with finished compost may also be done to ensure thermophilic temperatures reach the outer layer of the compost heap.

The main benefit of this technology is that it does not require any turning [55], which subsequently allows the compost heap to retain its heat effectively while still being able to supply the much-needed oxygen to the microorganisms via the passive aeration system. Consequently, the method may result in a slightly shorter composting period than conventional windrow composting. However, the absence of turning during the composting process necessitates more thorough preparations of the organic waste before putting it in the compost heaps.

### 4.1.3. Bin Composting

This technology is mainly practiced at a domestic household level with limited space, as depicted in Figure 4c. It can treat only a limited amount of waste and only produce compost for self-consumption. Organic wastes are commonly inserted from the top of a specially designed container with a perforated wall to allow convectional air flow to the compost heap. The organic composting material degrades and becomes compacted slowly as it gets down into the container, with the final ready compost collected from the bottom of the container. Some containers may also include a stirring mechanism to allow a convenient method of mixing the compost heap, and as such, improve the air-filled porosity of the heap. The use of the self-contained system with perforated walls allows heat retention while enabling air to be circulated throughout the compost heap. For more extensive composting operations, bin composters can also be used on a large scale by combining the passively aerated method with bin composting [56]. The technology requires medium funding and a minimal amount of maintenance. Additionally, it requires less space than windrow composters as the waste is piled up vertically in the bin. No turning is also required, with the exception of stirring, which may need to be performed occasionally [57]. However, the composting process may take longer than windrow composting as the waste is contained inside a bin, and no turning is performed.

### 4.1.4. In-Vessel Composting

In-vessel composting is a method that encloses the composting materials within a container or a vessel [58], as shown in Figure 4d. Installations vary from very high-tech options, with different parameters monitored to very low-tech alternatives. In all configurations, airflow and temperature can be more easily controlled using this technology via the air portals from the holes around and on the sides of the vessel, allowing some air to pass through, which speeds up the composting process. Turning takes place manually, and it needs to be turned more frequently during the first two weeks of the composting process to help with the aeration process as well as to control both temperature and moisture. Large batches of organic waste can be added and composted using this technology, with the main benefit of requiring less space than the previous technologies. The organic waste is reduced in volume, and usually, after three weeks to months, the compost is further treated in an open space for the curing stage. Additionally, less labor is required as the mixing or turning occurs within the vessel. However, it is capital-intensive and requires high maintenance, necessitating regular checks inside the vessel to ensure a favorable composting environment [59] and manual mechanical rotation.

### 4.1.5. Vermicomposting

Vermicomposting, as depicted in Figure 4e, is a type of composting in which microorganisms and certain macroorganisms, such as earthworm species, are utilized at room temperature to improve the organic waste decomposition process and to provide a better final compost [60]. The method is different from conventional aerobic composting, with specially chosen red worms, commonly *Eisenia Foetida,* added to the compost heap. These worms have high appetites and breeding abilities, can digest the organic waste materials and pass them through their digestive tract to produce vermicompost in the form of granules [61]. Essentially, vermicompost is the worms' feces, also known as castings, which are rich in nutrients. Their castings are packed with microbes, which help continue the decomposition process to produce the final compost. However, they need a comfortable space to live and work. Some bedding materials, either shredded paper or cardboard, have to be prepared inside a worm bin for the worms to live and work. They also need some moisture and organic waste. The timeline for the whole process varies depending on the quantity of worms, the temperature, and how much waste is added to the bin. Furthermore, worm reproduction can occur [62], which eventually floods up the worm bin with worms after some time, and this may require transfers to an additional worm bin to maintain effectiveness. Vermicomposting can decrease the pathogens in the process, albeit not as

effectively as traditional composting, as pathogens are generally eliminated quicker in hot conditions. However, the worms cannot survive very hot temperatures, allowing some pathogens and weeds to survive. This method requires relatively low costs, maintenance, and space [63]. The worms used have the ability to consume the organic matter quickly, resulting in a faster composting process with additional help from the microbes in their castings, and the method requires very little labor.

### 4.2. Automatic Composting Technology

There are various reasons for the relatively low adoption level of composting, including a lack of awareness, the relatively long time required for composting to complete, and lack of knowledge of the biological composting process. Manual composting methods require some time to produce good compost, in addition to monitoring and effort, which can be a hassle for busy working people. Additionally, some composting methods also require plenty of space [31], and hence, may not be suitable for those who live in urbanized areas. An automatic composter aims to solve some of these problems.

#### 4.2.1. Forced Aerated Windrow Composting

In a forced aerated windrow setting, blowers are installed at the end of perforated pipes to force airflows to the compost heaps, as shown in Figure 5a [64]. The blowers inject air into the compost heaps, especially during the active stage, to supply the much-needed oxygen for the microorganisms, and hence, allow decomposition of organic waste materials. Airflow can be adjusted by changing the frequency and duration of the blower. The compost heaps are also commonly insulated to prevent heat loss and allow thermophilic temperature throughout the compost heaps, including the outer layers. Due to the efficient retention of heat and the ability to supply oxygen without turning, the composting process is commonly shorter. Little labor is required as compost heaps need not be turned [65]. However, the method requires high investment, given the need for blowers and aeration channels for airflow. Maintenance is also high and requires high space requirements.

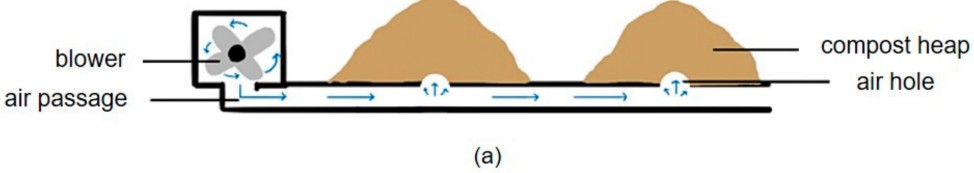

(a)

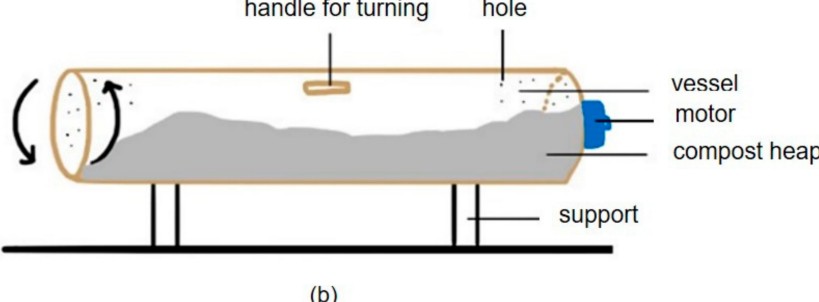

(b)

**Figure 5.** (**a**) Forced aerated windrow composting; (**b**) automatic turning in-vessel composting.

#### 4.2.2. Automatic Turning In-Vessel Composting

The technology illustrated in Figure 5b is similar to the manual in-vessel composting; however, the technology may vary in designs, size, and equipment. One of the automatic in-vessel composting processes is the motorized turning in-vessel composter. The automatic turning in-vessel composting uses a powered motor to rotate the vessels for aerating the compost heaps and can be scheduled to rotate at desired times and frequency [66]. Some

in-vessel composting processes also utilize a motor for rotating the vessel, and are equipped with temperature and humidity sensors to automatically monitor important parameters; replacing most of the manpower required in manual in-vessel composting. As a high torque motor is needed to rotate the heavy waste-filled vessels, the method requires very high investment and maintenance to ensure every piece of equipment works well to carry out the work. However, the in-vessel composting method is considered space-efficient and requires a low amount of labor.

### 4.2.3. Electrical Home Composter

An electric composter is an indoor compost bin that uses aeration, heat, and pulverization to minimize the volume, emissions, and odor of food waste. A common home composter may be small enough to fit on a counter, but for other types, large amounts of garbage also do exist and can be used in both indoor or outdoor applications. A home electrical composter, also known as a food recycler, uses three-phase cycles to break down food waste [67], with most composters taking an average of 24 h to a few days to break down waste into simpler compositions.

An electrical home composter attempts to provide the optimum composting environment occurring during selected phases of the natural composting process. Mesophilic and thermophilic phases are shortened in most electrical home composters through the applications of heat, such that excess moisture in the organic materials can be rapidly reduced. Organic materials are commonly automatically ground to increase their surface area to speed up the composting process before the materials are mixed with soils or additives to populate the microorganisms. After going through the accelerated mesophilic and thermophilic phases, the end products are eventually cooled down to room temperature to give entirely dry and sterile products, instead of the common texture of mature compost. This is because the end product from the home composter has only undergone partial phases of the composting process: the mesophilic and thermophilic phases, but has not undergone the maturation phase. These end products may be further cured outside of the system to ensure the resultant compost is adequately matured.

Electrical home composters may differ in terms of the adopted processes, quality of end product, and duration to completion. Nevertheless, the majority of the electrical home composters are based on three-phase cycles, which include drying, grinding, and cooling phases [68]. Some electrical home composters may also produce non-dehydrated and non-dry compost. This is possible due to the implementation of an additional phase, called the curing phase. In this phase, the organic materials that have been broken down into smaller substances are stabilized, applied with some heat, aerated, and turned until the compost is partially-cured (albeit not fully cured and stabilized) and able to be used as a garden compost as a final product. Due to the extra phase, these types of composters may take up to two weeks to complete the whole process. Table 2 shows the process description in each phase cycle.

**Table 2.** Process descriptions of each phase cycle in an electrical home composter [68].

| Phase Cycle | Process |
| --- | --- |
| Drying | Food recyclers attain an interior temperature of roughly 70 °C during the first drying phase to reflect the naturally occurring and ideal heat of a compost heap. The heat and aeration are evenly dispersed by the unit's grinding gears, which gently turn so that every surface area is disinfected and methane-free. Air is pumped through carbon filters and discharged out the back of the machine to supply air. The drying process reduces the volume of the initial organic materials. |
| Grinding | The unit's internal grinding gears then turn the contents once the food waste has been reduced in size. This further breaks down the food waste into minute, powder-like particles that may be easily mixed in with soil. |
| Cooling | This phase brings the unit and the contents of the bucket back to room temperature, allowing for safe handling. This phase also continues the previous phases' aeration and dehumidification. |
| Curing | Continuous aeration and moisture are regulated where the contents are allowed to stabilize for weeks. The curing phase cycle often takes longer than other phase cycles. |

There are quite a number of electrical home composters available in the market, with some of the products shown in Table 3 below.

**Table 3.** List of some available electrical home composters in the market.

| Product Name | Product Image | Features | Price in USD | References |
|---|---|---|---|---|
| Vitamix FoodCycler FC-50 |  | Size: 12.6″ × 11″ × 14.2″<br>Weight: 27 lbs<br>Capacity: 2.5-L bucket<br>Power consumption: 0.8 kWh/cycle<br>Processing time: dehydrated, ground, and cooled material in 4–8 h.<br>Phase cycle: drying, grinding, and cooling only | $400 | [69] |
| BeyondGREEN Composter |  | Size: 20″ × 12″ 20″<br>Weight: 22 lbs<br>Capacity: 5 lbs per day<br>Processing time: Compostable material in 5 days and high-nitrogen compost in 2 weeks.<br>Phase cycle: drying, grinding, cooling, and curing | $380 | [70] |
| Oklin GG-02 Composter |  | Size: 30″ × 18″ × 18″<br>Weight: 60 lbs<br>Capacity: 8 lbs<br>Power consumption: 60–90 kWh/month<br>Processing time: usable soil amendment in 24 h.<br>Phase cycle: drying, grinding, and cooling | $1200 | [71] |
| Lomi Composter |  | Size: 16″ × 12″ × 13″<br>Capacity: 7 lbs<br>Power consumption: 1 kW/h<br>Processing time: dry material in 20 h<br>Phase cycle: drying, grinding, and cooling only | $499 | [72] |
| KALEA Composter |  | Size: 9″ × 25″ × 20″<br>Capacity: 7 lbs<br>Power consumption: 200 kW/year<br>Processing time: 48 h into nourishing compost<br>Phase cycle: drying, grinding, cooling, and curing | $800 | [73] |
| NatureMill ULTRA Composter |  | Size: 20.3″ × 20″ × 12.6″<br>Weight: 25.4 lbs<br>Capacity: 120 lbs/month<br>Power consumption: 5 kWh/month<br>Processing time: compost in 2 weeks<br>Phase cycle: drying, grinding, cooling, and curing | $500 | [74] |

Electrical home composters are commonly embedded with different sensors for monitoring and control purposes. Temperature and moisture sensors are the most common sensors integrated onto an electrical composting system [75] to aid in the monitoring of thermal conditions inside the electrical composter and to ensure effective decomposition of the organic matter, preventing too wet or too dry contents, as well as to regulate the temperature from exceeding 70 °C, which can kill the microbes. An air pump is commonly

used as an actuator to regulate temperature, such that the temperature within the system does not rise beyond a certain value, which may kill the microorganisms. Grinders are commonly used to shred organic materials to increase the surface area for the microorganisms to act on. These systems can be commonly operated with just a click of a button to activate the process, equipment, and sensors, from start to end product completion.

Most food recyclers use ventilation and heat to quickly break down food waste, much like a pile of regular compost. However, most of the end-product is completely dry, sterile, and immature [76], and hence, it cannot be considered proper compost. A home composter is primarily designed as an alternative for conventional composting; by reducing the hassle of managing compost heap and, as such, is suitable for people who want an odorless composting process with limited space to carry out conventional composting methods. This composter is designed to be used by anyone and may facilitate the reduction of food waste or organic waste from home. Despite its fast process, the initial investment of the machine itself can be high and requires high maintenance. However, its low space requirement and very low labor requirement make it an attractive alternative to conventional composting.

### 4.2.4. Comparative Summaries of the Manual and Automatic Composting Methods

Table 4 shows the comparative summary of manual and automatic composting methods. The literature indicates that many benefits may be obtained by adopting an automatic composting process; faster, easier, and more convenient composting, which may encourage composting at the source. Subsequently, this may reduce the need to transport the bulk of the waste to landfills and its associated problems. However, despite a large number of scientific pieces of research on the topic, none have specifically addressed the technological advancements in composting from the patent perspective. This is despite patents representing valuable information on the technology and may indicate directions of the composting technology. As such, there is a clear need to perform a systematic patent review on the technological advancements in the field.

**Table 4.** Comparative summary between some of the manual and automatic composting methods.

| Manual composting | | | | | |
| --- | --- | --- | --- | --- | --- |
| **Method of composting** | **Cost** | **Maintenance** | **Space requirement** | **Composting duration** | **Labor requirement** |
| Windrow | Very low | Very low | Very high | Moderate | Very high |
| Passively Aerated Windrow | Low | Low | Very high | Moderate | Low |
| In-Vessel | Very high | Very high | Moderate | Fast | Low |
| Bin | Moderate | Low | Moderate | Slow | Low |
| Vermicomposting | Very low | Low | Low | Fast | Very low |
| **Automatic composting** | | | | | |
| **Method of composting** | **Cost** | **Maintenance** | **Space requirement** | **Composting duration** | **Labor requirement** |
| Forced Aerated Windrow | High | High | High | Fast | Low |
| Automatic Turning In-Vessel | Very high | Very high | Moderate | Fast | Very low |
| Electric | High | High | Low | Very fast | Very low |

## 5. Methodology of Review

Patent landscaping intends to give a clear picture of the technological advancement of a given field and aids in the development and implementation of a long-term research and development plan that considers different aspects of the technology. In order to successfully examine and acquire a clear direction, a patent landscape needs to follow a systematic sequence of processes. The patent review process shown in Figure 6 has been carried out in accordance with the PRISMA statement [77]; following a three-step process including searching for related patents and refining, filtering, and categorizing patents and finally, performing an in-depth review on the remaining patents relevant to an electrical composter. In this paper, the Derwent Innovation database has been utilized as a patent searching tool. It is one of the most extensive databases for global patent searches, comprising more than

36 million patent families derived from more than 100 million patents worldwide, which have been sourced from 50 diverse sources throughout the world [78,79].

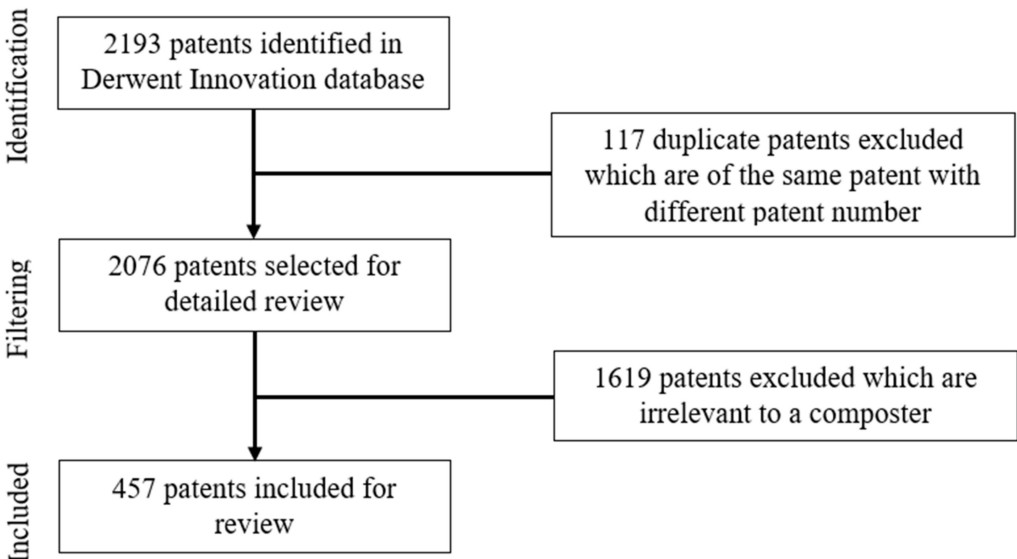

**Figure 6.** Patent review process in accordance with the PRISMA statement [77].

## 5.1. Patent Review Process

The patent search was conducted on the 30th November 2021, focusing on patents from the 1990s to the present, at the initial stage of the search process. Non-English patents, translated into English in the Derwent Innovation database, were also considered. A combination keyword and International Patent Code (IPC) search was used to identify relevant patents to the topic of an electrical composter. The IPC codes are generalized according to class groups, and it was observed that groups A, B, C, and F are most relevant to composting technologies. On the other hand, the keyword search considers composting technology and process, by focusing on the title, abstract, and claims of the patent documents. The keywords used in the patent search and descriptions of the IPC group are shown in Tables 5 and 6, respectively.

**Table 5.** Keywords used in the patent search.

| Category | Keyword |
| --- | --- |
| Technology | Search (Automatic composter or Composting Device or Waste Composter or Food Recycler or Compost Heater) (Title, Abstract, Claims) |
| Process | Search (Compost Collecting or Compost Heating or Aerobic Composting or Mixing Compost or Food Recycling or Organic Waste Treatment) (Title, Abstract, Claims) |
| IPC Group | Search (C05F or C12M or B02C or B07B or B65F or A23L) |

**Table 6.** Descriptions of selected IPC codes used in the search query.

| IPC Code | Description |
| --- | --- |
| A23L | Foods, foodstuffs, or non-alcoholic beverages, their preparation or treatment; preservation of foods or foodstuffs, in general |
| B02C | Crushing, pulverizing, or disintegrating in general; milling grain |
| B07B | Separating solids from solids by sieving, screening, sifting, or by using gas currents; separating by other dry methods applicable to bulk material |
| B65F | Gathering or removal of domestic or like refuse |
| C05F | Organic fertilizers, e.g., fertilizers from waste or refuse |
| C12M | Apparatus for enzymology or microbiology |

An initial patent search on the Derwent Innovation database with the combination keyword and IPC search gave an initial total of 2193 patents, with only published patents between the years 2000 and 2021 considered. The search results were refined further based on the technical criteria relevant to the search and further cleaned up to generate more accurate outputs, which are representative of the target technological area. A total of 117 duplicate patents, composed of patents from the same patent family, were removed. Titles, abstracts, and claims of the patents were also extracted and thoroughly analyzed to assess the patents' relevance to electric composters, with a total of 1619 patents removed as they were deemed irrelevant. Collectively, 1736 patents (79.2% of the total) were removed from the initial search results, and only 457 patents (20.8% of the total) were retained for further analysis.

It was noted that the patent review process was limited to patents available in the Derwent Innovation database only. Additionally, patents that may be related to electrical composter but have not been captured by the combined keyword and IPC search strategy were not considered for the analysis.

### 5.2. Patent Analysis

The next stage in the patent landscaping process was to evaluate the data thoroughly to categorize the results into more manageable and meaningful category types, as shown in Figure 7.

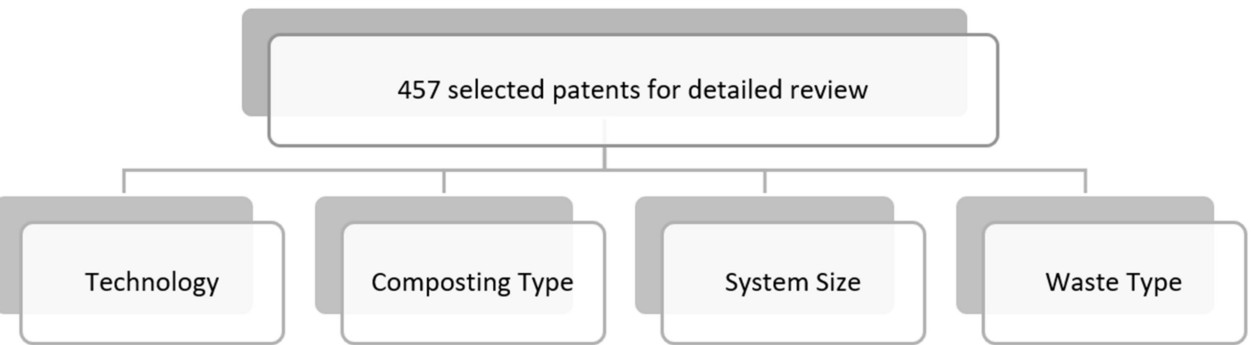

**Figure 7.** Categorization type of the selected patents.

The 457 patents were classified into four category types: technology, composting type, system size, and waste type (Figure 7). Table 7 gives the composition of patents in each category type and the descriptions of each category type considered in this paper. Some patents may fit into multiple categories. The patent categorization functions to reveal useful hierarchical categorization, enabling crucial and valuable insights for analysis from multiple perspectives.

There are three different categories of composting technology: manual, partially automatic, and fully automatic. The first technology category is manual, where the composting operation requires human intervention to monitor and carry out the required processes. The partially automatic category refers to a semi-automated composting operation that provides some automation, but at the same time requires manual human interventions for some operations. The last technology category is fully automatic, where the composting operation totally eliminates human intervention, with sensors used to monitor the composting processes and actuators to provide the necessary feedback.

In terms of composting process type, the patents are categorized as aerobic and anaerobic processes and an 'undetermined' category if the patent does not explicitly specify the composting process the invention is tackling. The aerobic composting process requires the presence of oxygen for composting to occur, while the anaerobic composting process occurs in the absence of oxygen.

**Table 7.** Category type descriptions and composition of patents in each category.

| Category Type | Description | Category | Composition of Each Category |
|---|---|---|---|
| Technology | Composting technology is exclusive where it can be either manual, partially automatic, or fully automatic | Manual | 22.76% |
| | | Partially automatic | 17.29% |
| | | Fully automatic | 59.96% |
| Composting process | Composting type is non-exclusive, where a patent may fit into multiple categories: aerobic, anaerobic, or undetermined. Undetermined refers to an unspecified type of composting in a particular patent | Aerobic | 86.50% |
| | | Anaerobic | 5.49% |
| | | Undetermined | 8.02% |
| System size | System size is exclusive where it can be either small- or large-sized | Small-sized | 26.91% |
| | | Large-sized | 73.09% |
| Waste type | Waste type is unexclusive where it can be at least one from the garden, agricultural, animal manure, food/kitchen, and general organic wastes (which can consist of the previously mentioned type of wastes) | Garden waste | 4.90% |
| | | Agricultural waste | 4.05% |
| | | Animal manure | 5.97% |
| | | Food/kitchen waste | 26.87% |
| | | General organic waste | 58.21% |

Meanwhile, two categories of system size type are considered: small-sized and large-sized systems. A small-sized composter refers to either a home composter that is small enough to be put on a kitchen countertop or a slightly larger-sized composter that can be placed outdoors, but still within the house compound. Having a small-sized composter means that it can only handle a handful of kitchen and garden waste. On the other hand, a large-sized composter can handle large volumes of organic wastes on a regular basis, and it is specially designed for industrial applications. Additionally, it also requires a spacious area and is often built within the waste management plant.

The waste category type defines the types of wastes that the patents are designed to deal with. These can be food waste, animal manure, garden waste, agricultural waste, and general organic waste. Food waste refers to leftover and discarded food, and animal manure refers to solid or slurry feces produced by some animals. Garden waste refers to the accumulated plant matter resulting from gardening activities, including trimmed grasses, weed eradication, and leaf debris. On the other hand, agricultural waste is unwanted waste generated as a result of agricultural activities, including crop residues, sawdust, and forest waste. General organic waste can be any of the previously mentioned waste types, with the patent not specific for the type of waste the invention is able to deal with.

*5.3. Detailed Review of Selected Patents*

The final stage of a patent review process includes an in-depth review of selected patents based on the composting technology category type: manually operated, partially automated, and fully automated technologies. Different patents on aerobic, anaerobic, or combined aerobic/anaerobic composting types within each technology category have been reviewed to summarize the novel inventions and the processes involved in the patents. This also provides a deeper technical understanding of the advancement of an electrical composter.

**6. Results and Discussions**

*6.1. Patent Landscape Overview*

The initial patent landscape search using the combined keywords and IPC search gave a total of 2193 published patents, which was reduced to 457 patents after accounting for duplicates and irrelevant patents. Only patent applications between the years 2000 and 2021 have been considered in this study.

Figure 8 shows the number of patents published between the years 2000 and 2021 by the top-five countries: China, Korea, the United States, Japan, and Canada, which have filed the most patents in the area. It can be seen that China filed the largest number of patents with a total of 278 patents or 60.83% of the total patents considered, and this was followed by Korea, the United States, Japan, and Canada with 49 (10.72%), 23 (5.03%), 22 (4.81%), and 19 (4.16%) patents, respectively. Collectively, these five countries filed 391 patents, or 85.56% of the total patents considered.

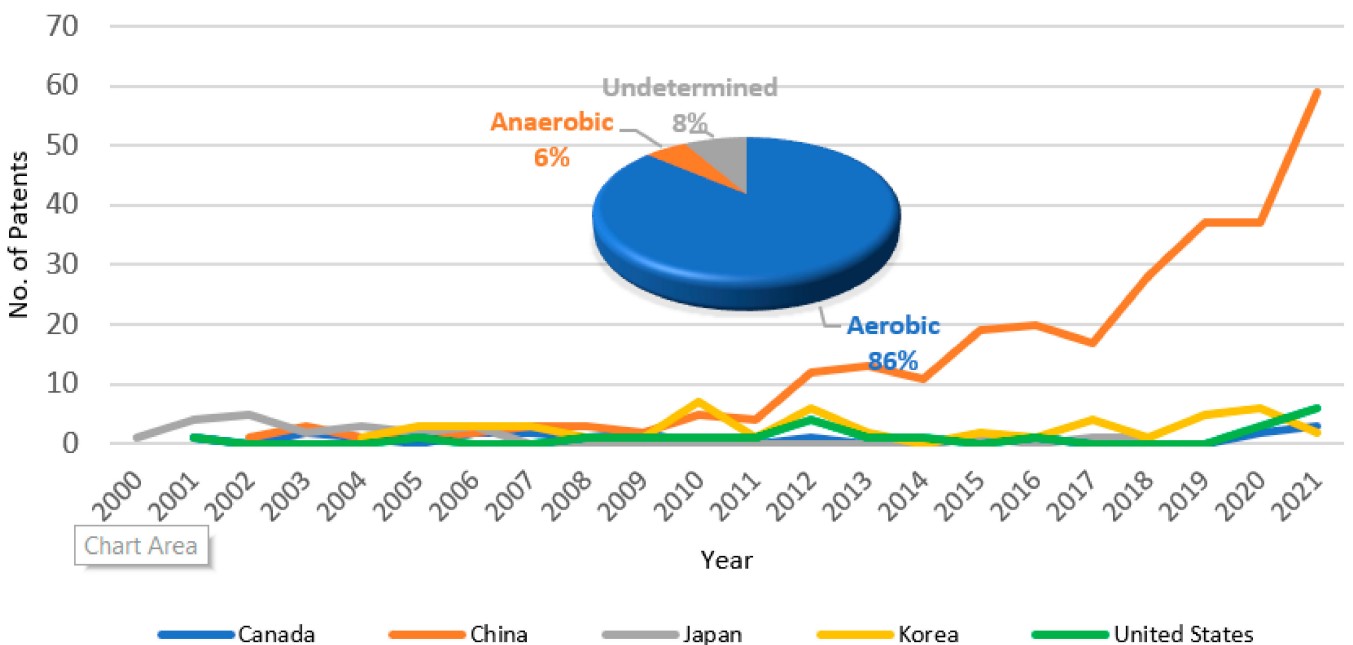

**Figure 8.** Patent activity by region and composting category types.

China became active in filing patents in the area as early as 2011, where it can be seen that China filed the highest number of patents compared to the other four countries. This interesting observation may be due to an increase in livestock production, amounting to an annual total of 3060 million tons in 2005 [80], which has led to an increase in manure production that needs to be processed. Additionally, it has been observed that there has been a change in policy by the government of China to encourage researchers to start filing patents on their research outputs in the last decade.

For the past 21 years, the aerobic composting process has dominated patents on composting technology, accounting for the highest proportion with 86%, as shown in the pie chart in Figure 8. Anaerobic composting technology only accounts for 6%, while 8% of the patents have not specified their specific technology usage. It can be deduced that aerobic composting is the preferred method of composting. This is due to its ability to break down raw materials quickly, produce an odorless end product, and attain sufficient high temperature necessary for eliminating pathogen and weed, and hence, produce high-quality compost. On the other hand, anaerobic composting generates digestate and biogas, a mixture of methane and carbon dioxide gases which is 25 times more potent than carbon dioxide in trapping heat in the atmosphere, leading to rapid global warming. Also, achieving such a system requires the use of a specialized and expensive tank to contain the potent gases requiring high capital investment and are considerably less profitable compared to other low-cost alternatives, therefore limiting the development of the technology.

Figure 9 shows the types of waste by patenting countries and the respective compositions of the waste types. General organic waste (58% of the total waste type) is the most common type of waste addressed by the patents for electrical composters, followed by food

waste (27%), animal manure (6%), garden waste (5%), and finally agricultural waste (4%), as shown in the pie chart in Figure 9. China filed the highest number of patents related to general organic waste, followed by food waste. Similar trends are seen for Korea, the United States, Canada, and Others, where Others represents patents filed by countries other than the five main countries. Despite the fact that China has seen a surging increase in livestock production, as previously mentioned, only a small number of patents (4.5%) had specified animal manure waste as the target waste type. However, general organic waste does include animal manure, and hence, the patent may be useful for the processing of animal manure. This has been further verified by looking into the specific details of different Chinese patents, showing that the patents can indeed be used for animal manure waste.

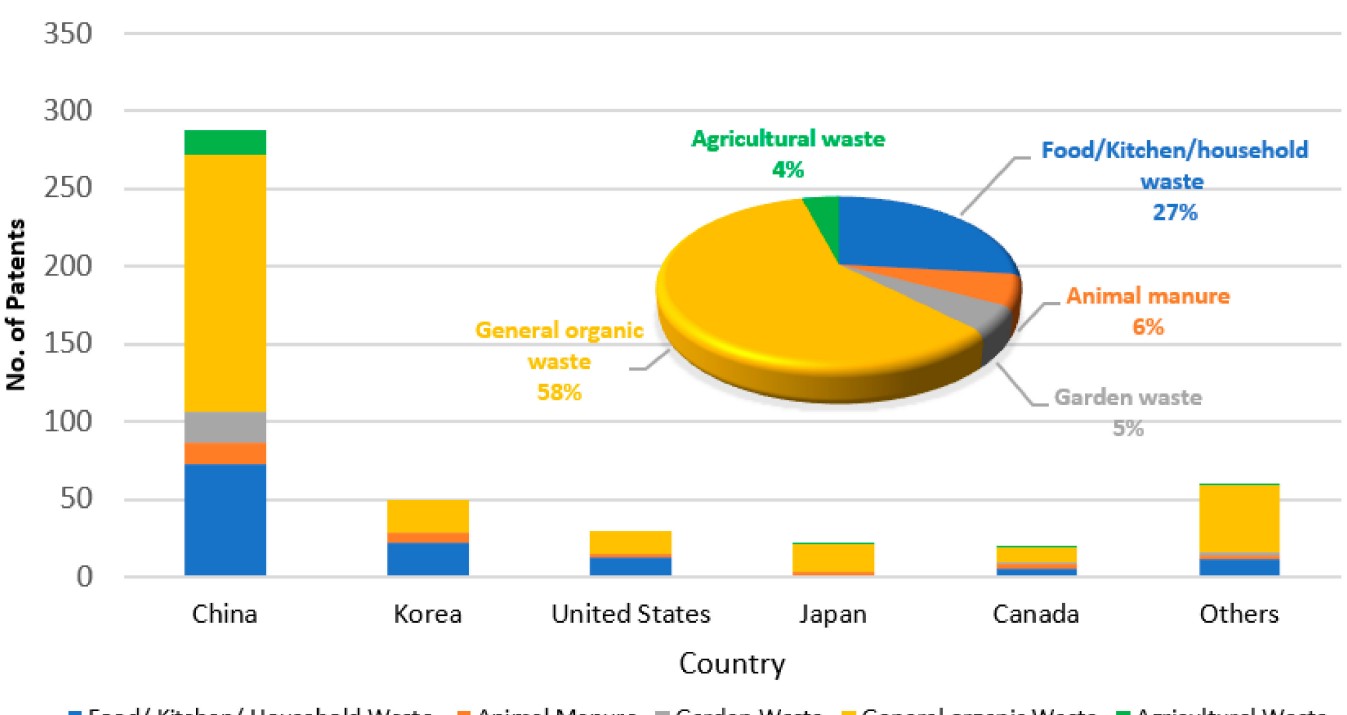

**Figure 9.** Types of waste by countries and the respective compositions of the waste types.

Figure 10 shows the categorization of the patents by waste types and system size. Patents intended for the processing of general organic waste (65% of the total patents considered for both system size), animal manure (7%), agricultural waste (6%), and garden waste (4%) are mostly designed as an industrial or large-sized electrical composter, with fewer patents designed as a household or small-sized electrical composter. The exception is for processing agricultural and food/kitchen/household wastes. Only large industry-scaled systems have been patented for inventions dealing with agricultural waste. Meanwhile, for food/kitchen/household waste type, it is interesting to observe that the proportion of the filed patents for both small- and large-sized electrical composters is almost similar, accounting for 64 and 62 patents, respectively. The vast number of patents designed as an industrial-sized system in proportion to a small-sized system indicates that composting is still primarily done on an industrial scale at a waste management center, as opposed to on a smaller scale. However, there is definitely interest in composting food/kitchen waste, as evident from 64 patents for food/kitchen/household waste, which have been designed on a small-size system.

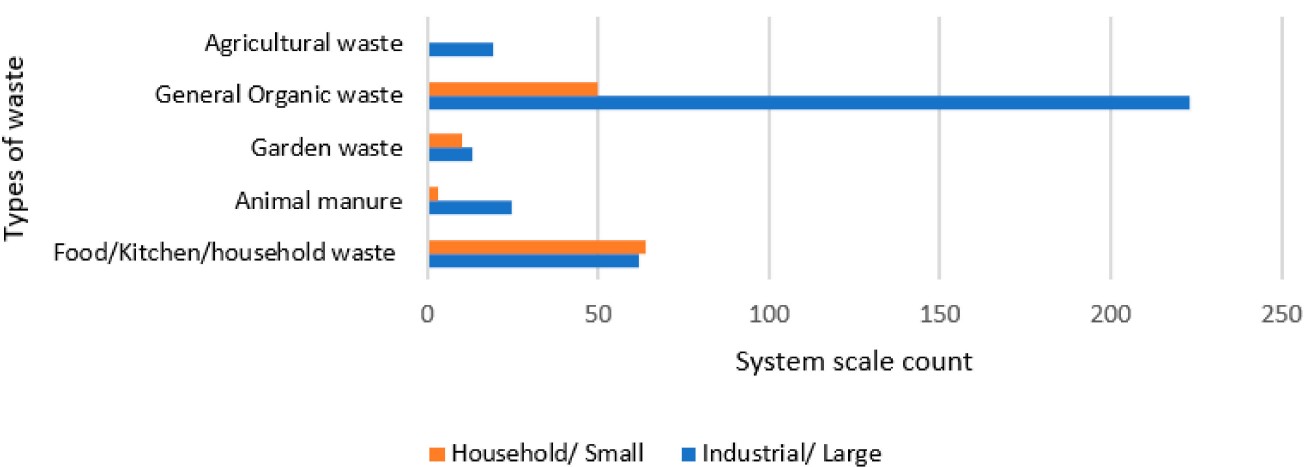

**Figure 10.** Waste types by system size.

It is noted that for an obviously large number of 273 patents, both system sizes had been categorized under general organic waste type. General organic waste type includes other waste types, with a majority of patents considered in this study not specifying the waste type categorically.

Figure 11 shows the frequency of the top-ten IPC group occurrences among the patents considered in this study. Descriptions of the top-ten IPC codes are shown in Table 8. The highest frequency IPC recorded is B09 with a total of 99 patents, followed by C05 with 88 patents, B01 with 65 patents, and B02 with 61 patents. It can also be seen that the majority of the filed patents with 392 patents from the total number of patents considered in this study belong to the IPC family B and C. This shows that most inventors are focusing on improving and innovating the operations of composting technology through improvision of the different composting processes.

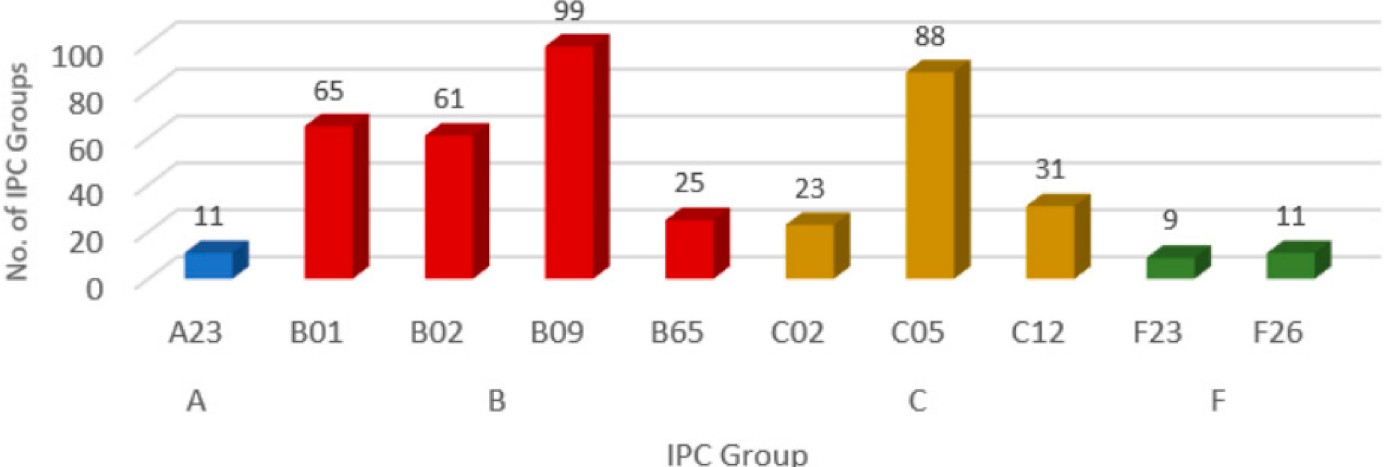

**Figure 11.** Frequency of top-ten IPC group occurrences.

**Table 8.** Details of the top-ten IPC code groups in patent analysis.

| IPC Code | Details |
|---|---|
| A23 | Foods or foodstuffs; treatment thereof, not covered by other classes |
| B01 | Physical or chemical processes or apparatus in general |
| B02 | Crushing, pulverizing, or disintegrating; preparatory treatment of grain for milling |
| B09 | Disposal of solid waste; reclamation of contaminated soil |
| B65 | Conveying; packing; storing; handling thin or filamentary material |
| C02 | Treatment of water, waste water, sewage, or sludge |
| C05 | Fertilizers; manufacture thereof |
| C12 | Biochemistry; beer; spirits; wine; vinegar; microbiology; enzymology; mutation or genetic engineering |
| F23 | Combustion apparatus; combustion processes |
| F26 | Drying |

Inter-relations between assignees and inventors have been constructed using a network diagram, as shown in Figure 12. Patent-based inventor network diagram illustrates the relationship between companies and inventors and helps visualize collaboration flows between and within companies. Many connections can be intimately linked to joint collaborations, thereby reducing duplicate research and minimizing resources simultaneously. Figure 12 shows three different clusters where joint collaborations between the same inventors and the same assignees have been observed.

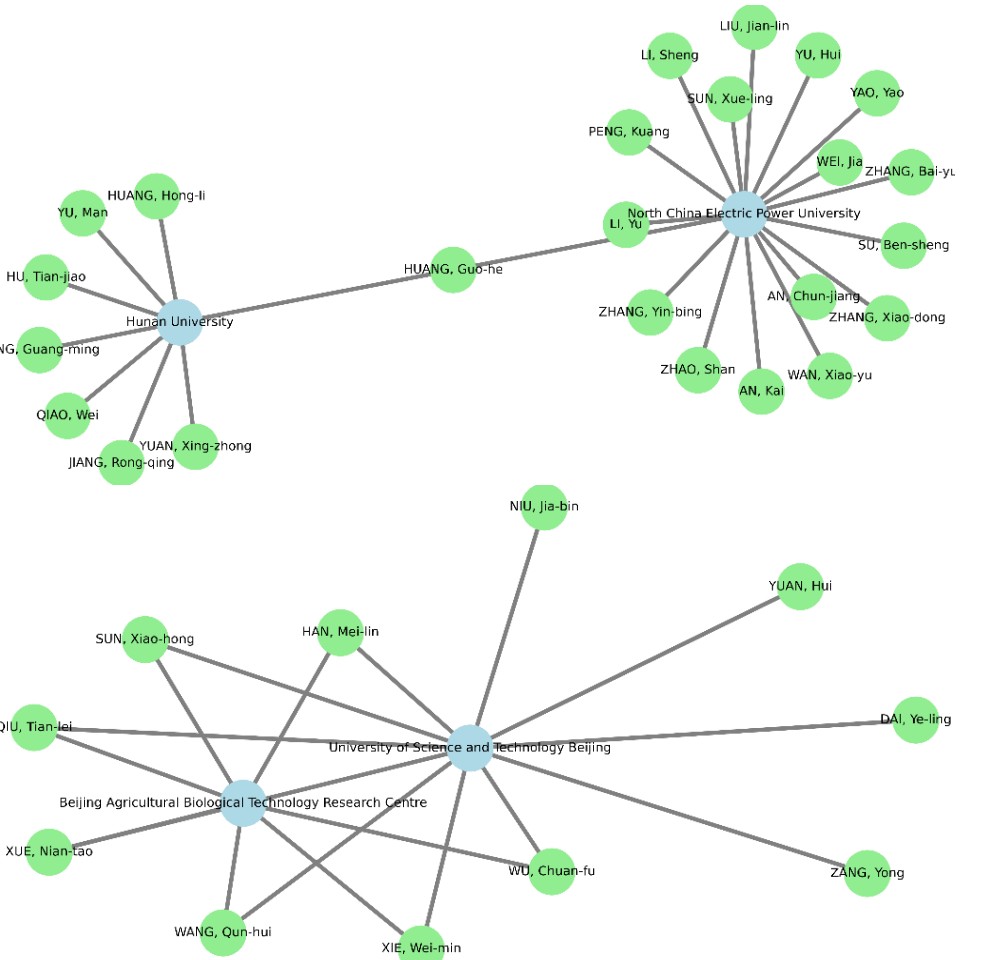

**Figure 12.** *Cont.*

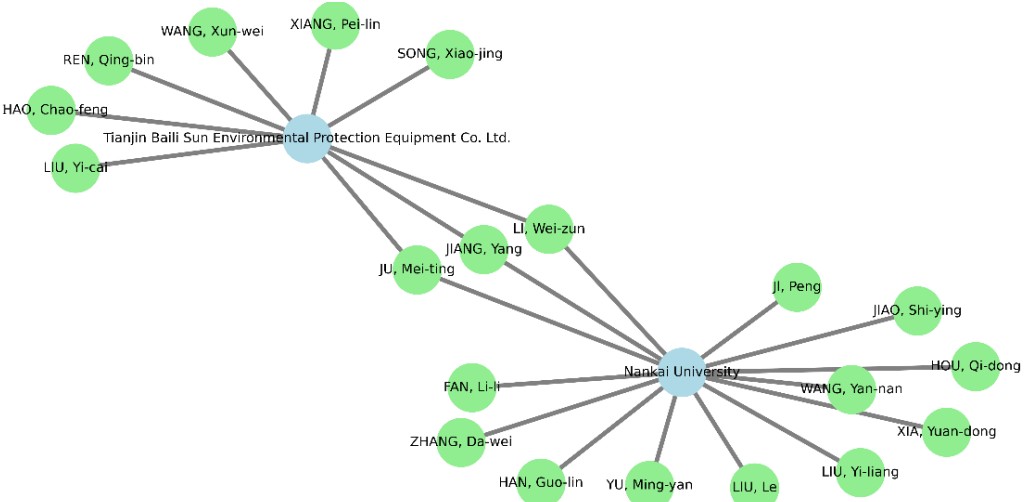

**Figure 12.** Network diagram of the assignees and inventors.

Inventor Huang Guo-He has tapped resources of two different universities; Hunan University and North China Electric Power University. Huang Guo-he filed one patent assigned to Hunan University in 2002 and filed two other patents in 2007, but assigned to North China Electric Power University. A year later, in 2008, he filed another two patents, assigned to Hunan University in April and North China Electric Power University in August of the same year. In 2011, he filed another one assigned by Hunan University. Meanwhile, inventors Wu Chuan-fu, Han Mei-lin, Xie Wei-min, Sun Xiao-hong, Wang Qun-hui, Qiu Tian-lei, and Xue Nian-tao had academic collaborations for similar research with the Beijing Agricultural Biological Technology Research Centre and University of Science and Technology Beijing on two different patents in 2009. Inventors Ju Mei-ting, Li Wei-jun, and Jiang Yang had industrial-academic collaborations between Nankai University and Tianjin Baili Sun Environmental Protection Equipment Co. Ltd. to produce one patent in 2014.

Figure 13 shows the overall composting technology trend between the years 2000 and 2021. In line with the industrial revolution 4.0, the commonly high labor requirement and time-consuming operations commonly associated with manual composting technology have been gradually replaced by partially automated and fully automated composting technology. There has been a steady increase in filed patents on fully automated composting technology, especially in the last four years. A total of 42 patents had been filed on fully automated composting technology in 2021, in contrast to partially automated and manual composting technologies, with only 14 and 20 patents filed, respectively. It is not surprising to observe the domination of fully automated technology over the past 21 years due to its faster-composting operations, it requiring less space, and minimal human intervention, making a fully-automatic electrical composter more preferable. This developmental progress in fully-automatic composting technology was made possible with the declining price of electronics and microprocessors.

Figure 14 shows the categorization of patents based on the composting process of the manual, partially automated, and fully automated composting technologies. Irrespective of technology types, it is evident that the aerobic composting process accounts for a large proportion of the composting process addressed by the patents, with 85, 90, and 86% for manual, partially automated, and fully automated technology, respectively. This is, of course, due to the faster aerobic composting process, which is capable of breaking down microorganisms quickly in the presence of oxygen, as compared to the anaerobic composting process.

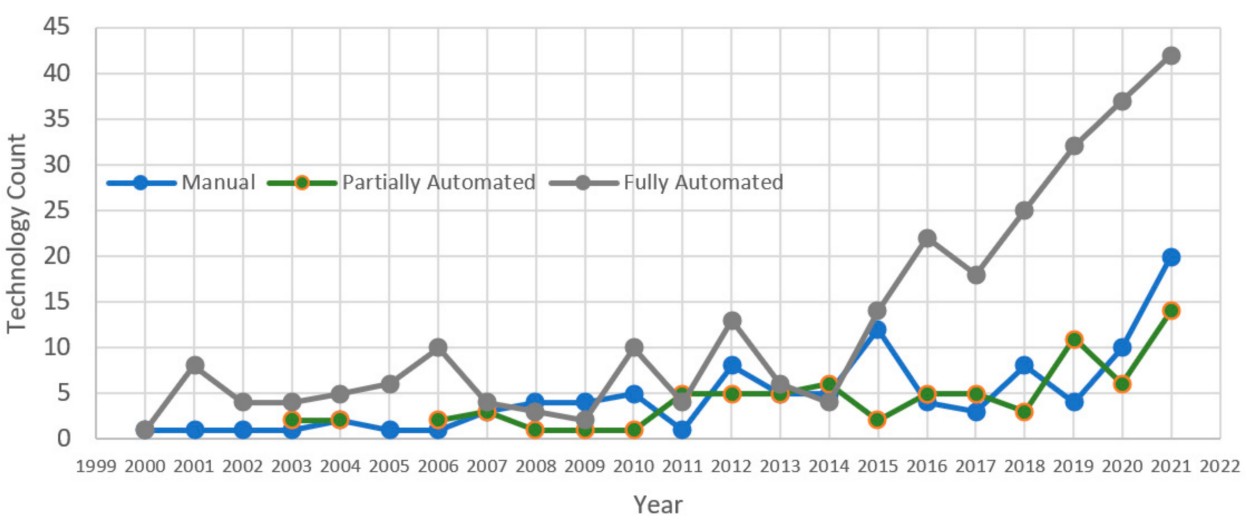

**Figure 13.** Technological trends of composting.

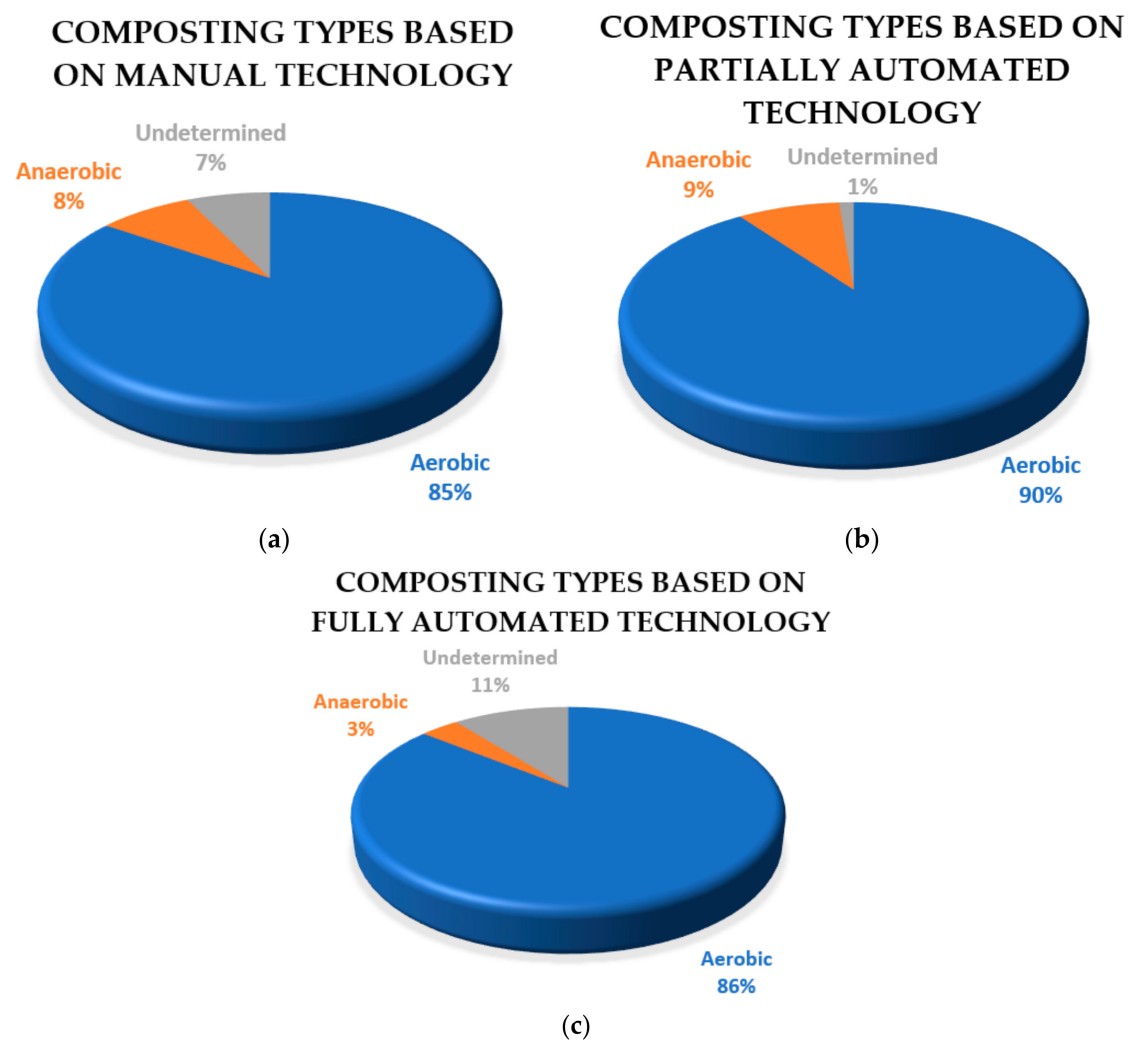

**Figure 14.** Composting process categorization of (**a**) manual technology; (**b**) partially automated technology; (**c**) fully automated technology.

Figure 15 shows the different processes addressed by the patents to achieve optimal composting conditions, for which the processes can either be chemically- or mechanically driven. The processes addressed by the various patents include drying, agitating, heating, aeration, filtering, cooling, maturation, and cleaning. The heating process introduces heat to the system, other than the natural heat due to the presence of microorganisms in a normal composting process, using a heater, or other heating mechanisms. It is noted that not all patents on composting technology consider all of the naturally occurring composting processes. A typical electrical composting system starts with the drying, shredding, heating, aeration, filtering, cooling, and finally, maturation processes.

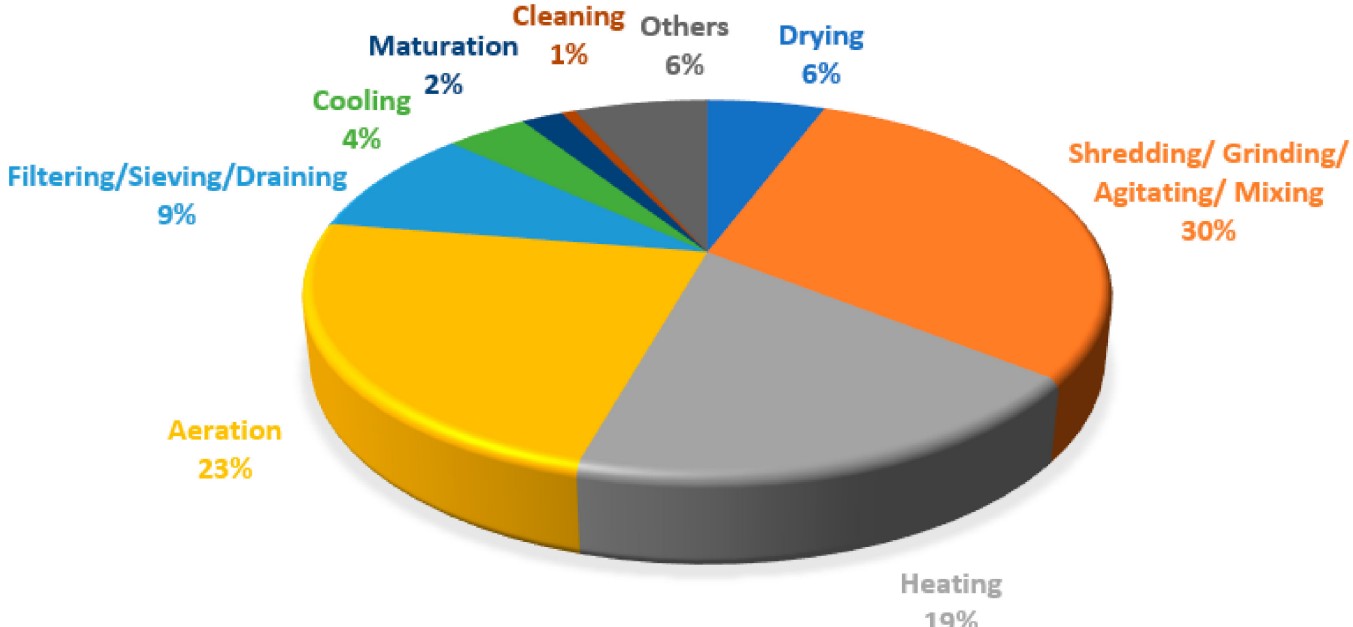

**Figure 15.** Processes involved for the selected patents.

Six percent of the total patents consider the drying process to remove excess moisture from the waste, while 30% of the total patents consider the shredding process. The heating process is considered in 19% of the total patents, with external heat applied artificially using a heater to supplement the naturally induced heat due to respiration and activity carried out by the microorganisms. The aeration process is considered in 23% of the total patents by artificially supplying oxygen to the compost materials, and 9% of the total patents consider the filtering process. A smaller proportion of 4 and 2% of the total patents consider the cooling and maturation processes, respectively. Other types of processes that are less common in composting, such as green energy harvesting to power the electrical composters, are categorized as Others. It can be seen that shredding or mixing, aeration, and heating are considered in the largest number of patents filed, as they are deemed to be significant to the composting process and hence, require control.

With the implementation of automated technology, taking accurate and reliable measurements of process parameters is undeniably vital towards optimization of the composting process. Therefore, further analysis on sensors involved within the selected patents has been carried out. For partially and fully automated technologies, temperature, moisture, oxygen, weight, and pH sensors have been utilized in the patents to reduce manpower requirements and monitor the composting conditions. Figure 16 shows the different types of sensors implemented in the electrical composters, with the majority of the patents focusing on temperature sensors (40% of the total sensors). Temperature is crucial in the composting process as different microorganisms are responsible at different phases of com-

posting and are temperature-sensitive. Therefore, temperature must be monitored closely; hence, temperature sensors account for the highest proportion of sensors considered by the patents compared to the other sensors. This is followed by moisture and oxygen sensors with 19 and 16% of the total sensors, respectively.

**Figure 16.** Sensors involved within the selected patents.

Control feedbacks are equally important in an automatic composting process, with the control feedbacks provided by actuators to provide an optimal environmental condition for the composting process. The implementation of control feedback necessitates the monitoring of at least one parameter using sensors. The controls may involve any one or multiple categories, including C:N ratio, particle size of compost, oxygen concentration, moisture concentration, temperature controls, and Others. The control feedback is categorized as Others if it is only considered in a few patents, such as controlling the emission of tail gases produced from the composting process. Figure 17 shows the different control feedbacks involved in the selected patents. As mentioned earlier, control feedback commonly responds to sensor outputs, and hence, it is not surprising to observe that temperature control accounts for 40% of the total control feedback considered by the patents due to the importance of operating within the optimal temperature range during the composting process. Operating outside the optimal range can pose some unfavorable effects leading to an ineffective composting. It is also observed that after temperature control, moisture control and oxygen control represent other control feedbacks that are addressed by many of the patents (21 and 18% of the selected patents in the review, respectively). This is despite that controlling temperature inadvertently affects the moisture and oxygen levels of the compost.

# CONTROL FEEDBACKS INVOLVED

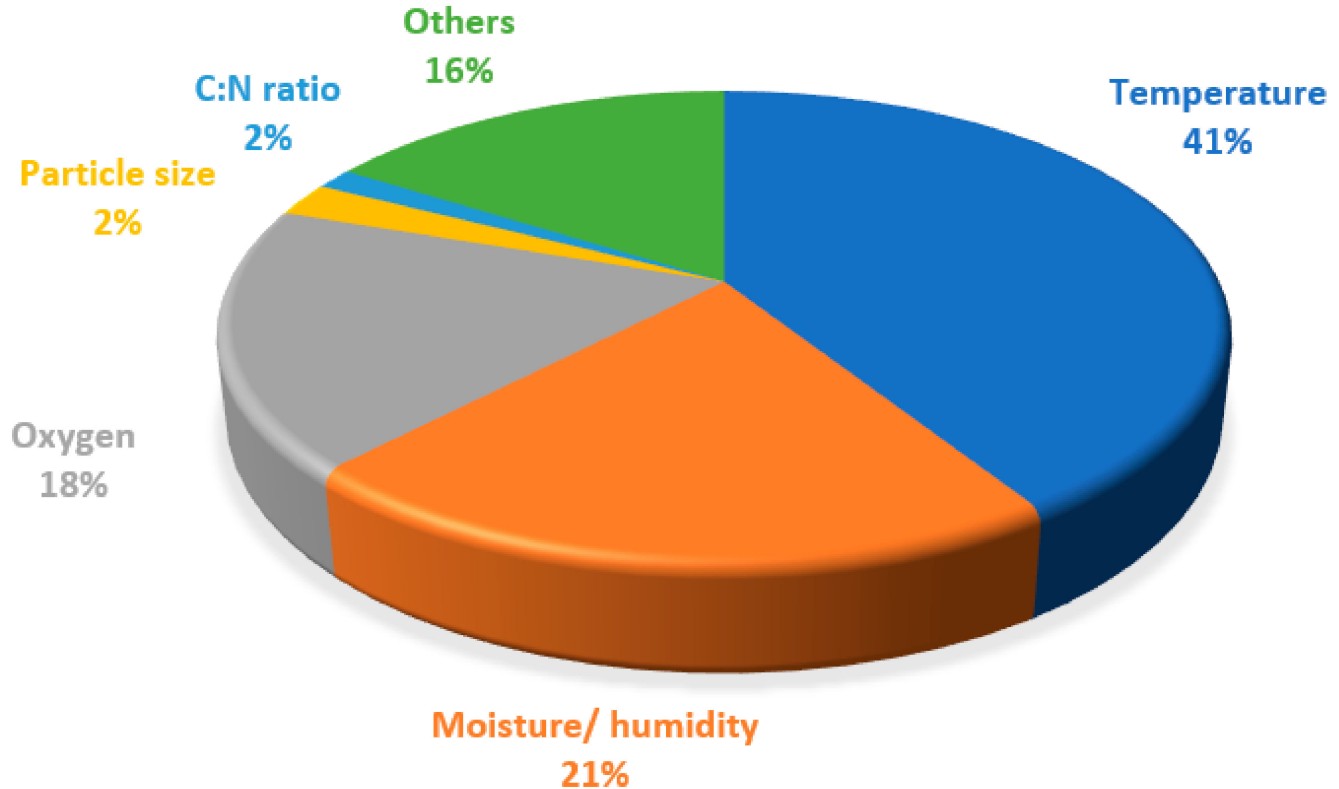

**Figure 17.** Control feedbacks involved within the selected patents.

## 6.2. Technology Updates

This section discusses in detail the technological updates of selected patents on manual, partially automated, and fully automated composting technologies, which have been further categorized in terms of process types: aerobic, anaerobic, and undetermined. Studies on the novel inventions and processes involved on carefully selected patents are provided to give insights into the development of the technology.

### 6.2.1. Technological Updates on Manual Composting Technology

Technological reviews on selected patents, focusing on manual composting technology with the three composting processes: aerobic, anaerobic, and undetermined, are summarized in Table 9 and discussed in-depth below.

A Canadian patent, CA2328680C [81], describes a unique aerobic composting tool for use in a receptacle (no. 4) to compost material by utilizing an aerator (no. 13) to ramp up oxygen flow through the contents of the receptacle, as seen in Figure 18a. Further details on the numberings can be viewed in Appendix A, Table A1. The invention seeks to alleviate oxygen-deprived issues during the composting process by incorporating an aerator to increase oxygen flow through the contents of a composting container. On the other hand, another Canadian patent, CA2671248C [82], innovates by designing a novel composter body (no. 12) with blow-molded plastic components' spaced apart walls (no. 38, 42, 40, and 56) and a hollow interior portion (no. 36), as shown in Figure 18b. Further details on the numberings can be viewed in Appendix A, Table A2. The design aims to insulate the composter, which helps trap heat within the composter and hence, minimize heat reduction from the composter itself. The insulating properties of the blow-molded plastic components help in heating and, at the same time, maintain an optimum temperature in the compost heap.

**Table 9.** Classification of the manual composting technology.

| Composting Type | Patent Number | Patent Title | Inventors |
|---|---|---|---|
| Aerobic | CA2328680C | Composting Device | Morrison Michael Joseph. |
| | CA2671248C | Composter | Stanford Carl R., Ashby Kent. |
| | CN103449849B | A kitchen waste compost device and composting method | Qu Xiao-lin, Wang Xue-jiao, Zhao Xue-fei. |
| | CN206872695U | A kitchen garbage composting device | Li Jing, Zhang Tian-zhu, Wang Shun-sheng, Guo Jing-jing, Wang Shuai, Guo Dun, Liu Jin-cheng. |
| | CN210163350U | Compost fermentation barrel | Chen Shi-Jiang. |
| | CN212051158U | Rotary drum-type composting device | Yuan Yu-Zhe. |
| | CN213977467U | Horizontal composting device | Guo Cong-jun, Meng Ying. |
| Anaerobic | CA2319808C | Improvement in composting toilet | Lejgren Harry. |
| Undetermined (Both Aerobic and Anaerobic) | CN204824645U | An organic garbage alternate aerobic and anaerobic composting | Zhou Shao-qi, YuanJin-peng, Yang Zhi-quan |
| | CN1206189C | Anaerobic and aerobic integrative type compost response operator | Qiao Wei, Zeng Guang-ming, Huang Guo-he, Yuan Xing-Zhong |

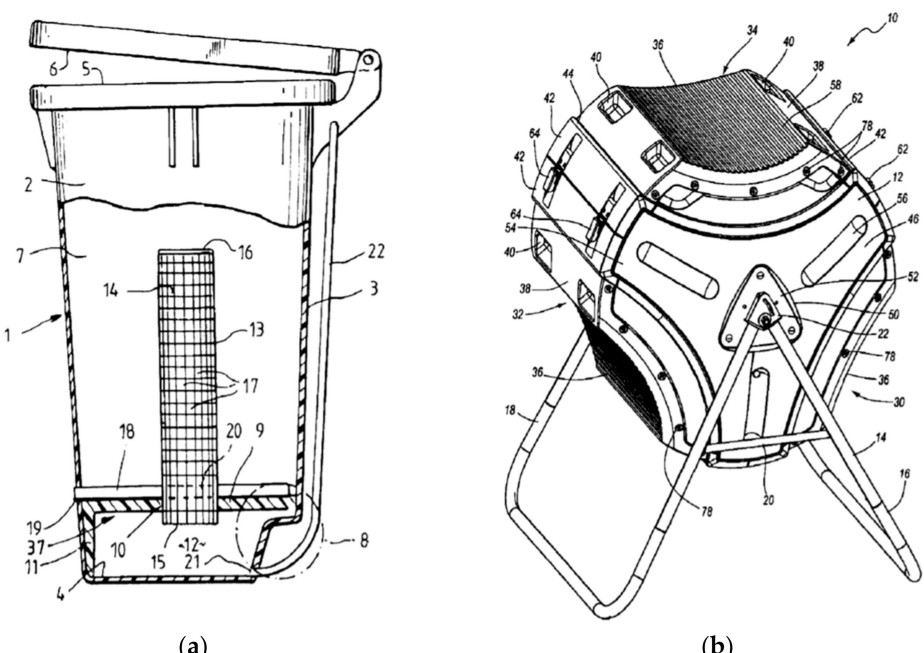

(**a**)  (**b**)

**Figure 18.** (**a**) An aerobic manual composter from CA2328680C [81], with details in Appendix A, Table A1; (**b**) an aerobic manual composter from CA2671248C [82], with details in Appendix A, Table A2.

A Chinese patent, CN103449849B [83], develops a new multi-functional kitchen waste composting device that is both practical and environmentally friendly in order to achieve on-site treatment and effective utilization of kitchen waste at the source. The composting device has been developed aesthetically and is both practical and ornamental to encourage composting. CN206872695U [84] describes a kitchen garbage composting device with a turning that rotates the blade within the device to crush the kitchen trash and evenly stir the viable bacteria in the kitchen waste. The kitchen waste is crushed to a size of 2–3 mm or less, and hence, increased surface area to improve the composting process. Another Chinese patent, CN210163350U [85], designs a composting barrel fermentation that consists of a barrel body with an internal heating and stirring device. The barrel body has a number of filtering holes to remove the liquid waste from the bottom of the barrel body via the

liquid outlet. Dry fertilizer is fermented to form compost within the barrel body, with the heating and stirring devices providing optimum temperature and aeration, respectively, to increase the fertilizer's fermentation efficiency.

Another Chinese patent, CN212051158U [86], develops a compact rotary drum-type composting device with a hand-held rotary wheel designed to easily rotate the waste and provide aeration. The device is made of a mesh plate heat preservation material, thereby providing sufficient oxygen supply, reducing energy consumption, and avoiding the formation of strange odors or leachate. CN213977467U [87] describes a horizontal composting device for collecting wet waste and fermenting it into compost. A liquid collecting tank connected to the liquid outlet pipe is placed at the bottom of the lower cover, and a stirring device is provided to mix the wet waste. The invention's technological solution has a stable structure, and is capable of stirring and speeding the wet-rubbish fermentation.

Categorized under manual technology patents with an anaerobic composting process, a Canadian patent, CA2319808C [88], describes a novel composting chamber, isolated from a detachable compost collection container by a perforated metal grid that allows filtering of compost particles. A rolling damper has been designed to control the moisture balance in the compost by regulating the degree of openness of the holes to ensure an effective decomposition process. It has been demonstrated that without the rolling damper, liquid may seep into the humus collection container, leading to over-wet compost and subsequently to slow down or stop the decomposition process. The liquid with the wet compost may also spill over, leading to dampness and foul odors.

Under the undetermined category, a Chinese patent, CN204824645U [89], describes a composting device that utilizes positive pressure ventilation to supply oxygen. The invention provides an alternative aerobic and anaerobic composting device for organic waste that can achieve consistent stirring of materials and supply an adequate supply of oxygen, which can significantly reduce the composting cycle of organic waste. Additionally, the device also allows maintenance of the reaction chamber during the organic waste composting process. This flexibility in maintenance prevents the emission of unwanted gases into the environment. Another Chinese patent, CN1206189C [90], designs a dual anaerobic and aerobic integrated reaction device for organic waste composting, consisting of a ventilation system (no. 8), a stirring system (no. 13), and a heating system (no. 2), as shown in Figure 19. A blower (no. 8) supplies oxygen into the device, with its blade (no. 14) constantly spinning fresh air into the waste material for the aerobic composting microorganisms to meet their oxygen demands. The invention also includes a circulating water bath heating jacket (no. 2) that maintains a temperature of 100 °C or less, allowing it to meet the temperature requirements of anaerobic waste digestion and aerobic composting. Further details on the numberings can be viewed in Appendix A, Table A3.

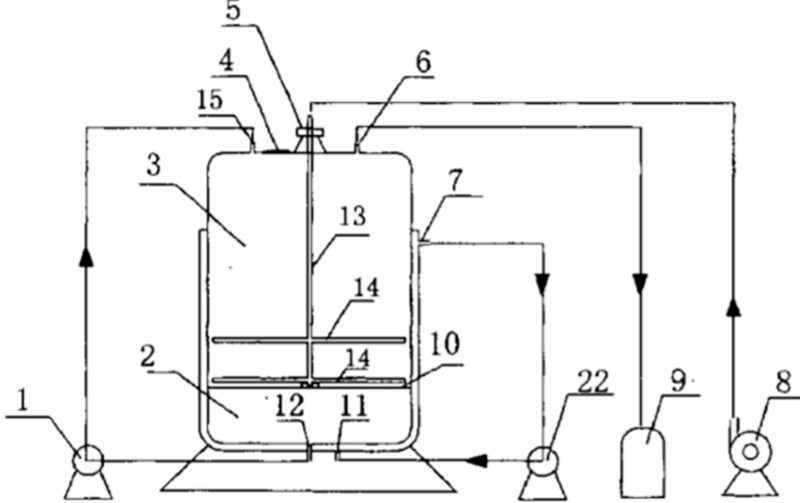

**Figure 19.** An anaerobic manual composter from CN1206189C [90], with details in Appendix A, Table A3.

6.2.2. Technological Updates on Partially Automated Composting Technology

Several technological reviews on selected patents focusing on partially automated composting technology with the three different composting processes: aerobic, anaerobic, and undetermined composting processes, are summarized in Table 10 and discussed in-depth below.

**Table 10.** Classification of the partially automated composting technology.

| Composting Type | Patent Number | Patent Title | Inventors |
|---|---|---|---|
| Aerobic | CA2436322C | Rotatable Aerating Composter | Windle Harry Neal |
| | CN108383556A | A fast, harmless organic waste processing method and system | Huang Bing-Feng |
| | KR2019021983A | Food Compost | Moon Jo Young |
| | US20120021504A1 | Aerated Composter and Waste Collection Bin | Bradlee Michael |
| | CN101983951B | Composting device for domestic waste | Li Bing, Dong Zhi-Ying, Chen Yu-hui, Zhu Jianlin, Cai Zhao-qi |
| | CN107021795A | A kitchen garbage composting device for family and composting treatment method | Xu Wei-ping, Deng Ying, Xu Jian-qiang |
| | CN205088151U | Aerobic composting device | Guo Chun-yu, Suo Ya-li, Wang Wei-dong |
| Anaerobic | AU2021204513A1 | Apparatus, Methods, and Systems for Food Waste Recycling | Boyle Norman |
| | CN112354616A | Food waste treatment device capable of recycling resources | Chen Ben-Zhong, Zhao De-long |
| Both Aerobic and Anaerobic | CN1248792C | Organic waste material treatment process | Rudas T |

A Canadian patent, CA2436322C [88], describes a system (no. 100) for an aerobic composting process. Further details on the numberings can be viewed in Appendix A, Table A4. The large-scale system, shown in Figure 20a, includes temperature monitoring, as well as watering and dewatering sub-systems. The invention ensures ample oxygen is available for an efficient aerobic action in an electric drive. A Chinese patent, CN108383556A [91], describes a method and system for quick and safe treatment of organic waste, which can reduce the composting treatment cycle and thus improve composting treatment efficiency, by adding organic waste with a water content of 60% into the treatment chamber. Additionally, a biological agent is added to the organic waste. The organic waste is agitated in the treatment chamber as well as heated to maintain an internal temperature of 75–85 °C. Agitation of the organic waste is performed for 0.5–2 h, and the contents are periodically disposed into the processing chamber while adding sufficient oxygen to allow for efficient decomposition. Meanwhile, a Korean patent, KR2019021983A [92], developed a compost maker to process food waste suitable for both small and large-scale systems. The novelty of the patent involves the inclusion of a pair of polygonal left-side and right-side faceplates with air holes in a square, hexagon, or octagon shape, formed with a rotation shaft through a hole at its center.

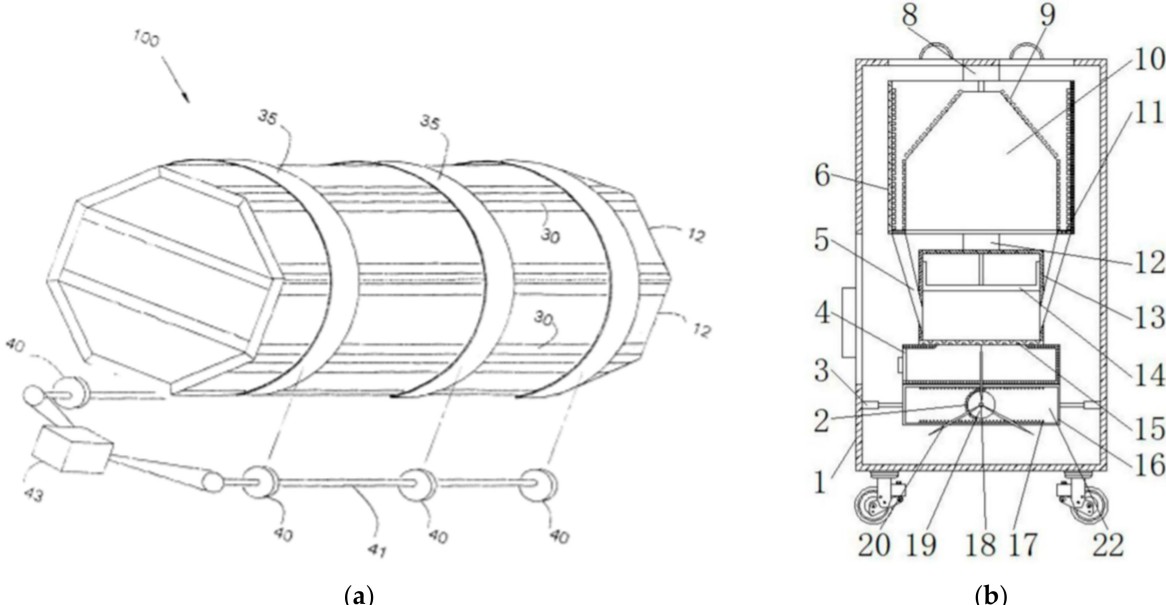

**Figure 20.** (**a**) An aerobic partially automated composter from CA2436322C [88], with details in Appendix A, Table A4; (**b**) an anaerobic partially automated composter CN112354616A [93], with details in Appendix A, Table A5.

Meanwhile, a United States patent, US20120021504A1 [94], describes a compost bin that can reduce odors, provides easy collection of compost tea from composting material, and is cost-effective, making the proposed system favorable in suburban and urban settings. It consists of a tube with holes connected with a vent tube and a drain to allow liquids to drain out. A Chinese patent, CN101983951B [95], describes a household waste composting device consisting of a ventilated helical stirring propeller. The device produces heat energy to accelerate the rate of fermentation and decomposition of kitchen waste and other organic household rubbish by creating a beneficial microbial activity environment and optimization of heat energy use rate. It employs solar heating, which is environmentally friendly and minimizes typical energy use.

Another Chinese patent, CN107021795A [96], describes a kitchen waste composting system and composting treatment procedures for the household. The system includes a control device to manage the heating insulating layer temperature and speed of the stirring device, to increase the stability of the compost product. CN205088151U [97] describes an aerobic composting system made up of a hollow pipe as its stirring device, with the surface of the stirring device consisting of multiple ventilation holes. The stirring device is connected to a blower. Consequently, the system can uniformly oxygenate and stir the compost, to effectively improve oxygen content in the compost.

Under the anaerobic composting process category, an Australian patent, AU2021204513A1 [98], describes an invention that simplifies food waste collection by automating some processes. A container has been designed to allow automatic emptying into a food waste collector, thereby simplifying the emptying process and reducing the manual effort normally required. Another patent, CN112354616A [93], develops a food waste processing device (no. 1) with crushing (no. 10) with the aid of a rotating motor shaft (no. 7 and 8), granulation (no. 11), and drying (no. 4), as shown in Figure 20b. Further details on the numberings can be viewed in Appendix A, Table A5. The food waste is crushed to a specified size and then filtered via a first sieve plate before being transported to the granulating box. Simultaneously, the rotations of the fan blade significantly improve the air convection rate of the drying box and improve the particle material drying forming efficiency.

A Chinese patent, CN1248792C [99], describes a device for treating organic waste materials that involve anaerobic digestion of the waste followed by aerobic composting of the residues in one vessel. The organic waste is treated by replacing the vessel's air and

contents with water, followed by the separation of gaseous by-products from the anaerobic digestion. A portion of the water in the vessel is then removed, and air is added to the remaining waste in the vessel to create aerobic composting conditions.

6.2.3. Technological Updates on Fully Automated Composting Technology

Several technological reviews on selected patents focusing on fully automated composting technology with the three different types of composting processes: aerobic, anaerobic, and both aerobic and anaerobic, are summarized in Table 11 and discussed in-depth below.

**Table 11.** Classification of the fully automated composting technology.

| Composting Type | Patent Number | Patent Title | Inventors |
|---|---|---|---|
| Aerobic | CN112960996A | Composting device based on a spiral stirring structure | Sun Guo-tao, Wang Xiu-zhang, Liu Xiao, Shao Zhi-Jiang |
| | CN208649153U | A household kitchen waste composter | Wu Han-zhang, Shen Kai-bin |
| | CN209555095U | An outdoor kitchen waste compost fermentation barrel | Liu Yu-ting, Zeng Xiang-lai, Xian Yi-nan, Chen Lai-yi |
| | IN201841044613A | A device for converting organic waste into organic compost | Giridhara Baluvaneralu Venkatakrishnaiah, Thazhe Vilippavil Muhammed Sameer |
| | CN111187101A | Household kitchen waste treatment device | Wang Mei-yin, Yang De-ming, Zhou Ting-jin, Meng Han-yu, Zhang Min-ling, Zhu Liang |
| | US8129177B2 | Automatic self-contained compost device | Cohn Russell S. |
| | US6284528B1 | Small scale automated composter | Wright James |
| Anaerobic | EP2980203A1 | Anaerobic digester for the treatment of organic waste | Arribas De Paz Ricardo |
| Both Aerobic and Anaerobic | CN212954904U | A new type of household garbage composting device with deodorizing function | Wang Yi-da |
| | CN208346058U | Organic garbage aerobic and anaerobic composting device | Zhang Xiao-hong |

Under the aerobic composting process category, a Chinese patent, CN112960996A [100], describes a spiral mixing composting device with a temperature sensor to monitor the temperature and an oxygen sensor to monitor the aeration. A motor is used to rotate the reaction barrel, and a blower feeds air into the reaction tank via the ventilation pipeline. It is claimed that the invention can enhance the aerobic effect, quickens compost maturity, prevents problems associated with uneven and inadequate mixing in a traditional composting mixer, controls the temperature of the composting body, and has a full response and high efficiency. Another patent, CN208649153U [101], describes a kitchen garbage composter. The composter is shown in Figure 21a and includes a filter screen (no. 41) and a liquid container (no. 42) set beneath the filter to separate the solid fertilizer from the liquid fertilizer effectively. Furthermore, infrared sensors (no. 161) monitor the opening and closing of the lid (no. 15) and liquid level sensor (no. 26) to determine the height of the liquid in the liquid waste container for monitoring purposes. Further details on the numberings can be viewed in Appendix A, Table A6.

In CN209555095U [102], an outdoor kitchen waste compost fermentation cylinder has been invented with the capability to insulate heat and prevent corrosion to the cylindrical structure, as can be seen in Figure 21b. The device is based on automation with the presence of an LED display screen (no. 23), a storage battery (no. 2), oxygen concentration detector (no. 8), which functions to monitor aeration, and infrared sensor (no. 19 and

22) to monitor the closing and opening of lid. Further details on the numberings can be viewed in Appendix A, Table A7. Meanwhile, an Indian patent, IN201841044613A [103], describes the production of organic compost in multiple stages by using three different chambers. Each stage receives the appropriate compost additive and is kept at the optimal temperature. The invention consists of rotatable chambers, a motor to rotate the chambers, and a hot air blower to initiate the thermophilic phase artificially, thereby accelerating the composting process.

CN111187101A [104] describes a household kitchen waste processing device consisting of a wastewater collection unit, a stirring mechanism, and a crushing device to crush the kitchen waste. Air is supplied to provide sufficient oxygen in the composting bin, with a filter orifice plate designed to collect waste water easily. The automatic device has been designed to facilitate the continuous treatment of kitchen waste at home. A United States patent, US8129177B2 [105], describes an automatic self-contained compost device that includes thermally insulated housing to allow the compost to reach temperatures suitable for quick composting, a motorized U-shaped mixing wand designed to mix the composting material, and a cure tray arranged below the reactor for receiving compost material dropped from the reactor. The device is automatic and requires almost no maintenance or cleaning. Another patent, US6284528B1 [106], describes a composting device with an airtight container to prevent smells from escaping out. A mixing mechanism is introduced to mix the materials, and airflow is directed downwards through the composting chamber. The benefits of this device include the elimination of manual compost turning, the ability to biodegrade animal waste, and the emission of non-noxious gas through the vent.

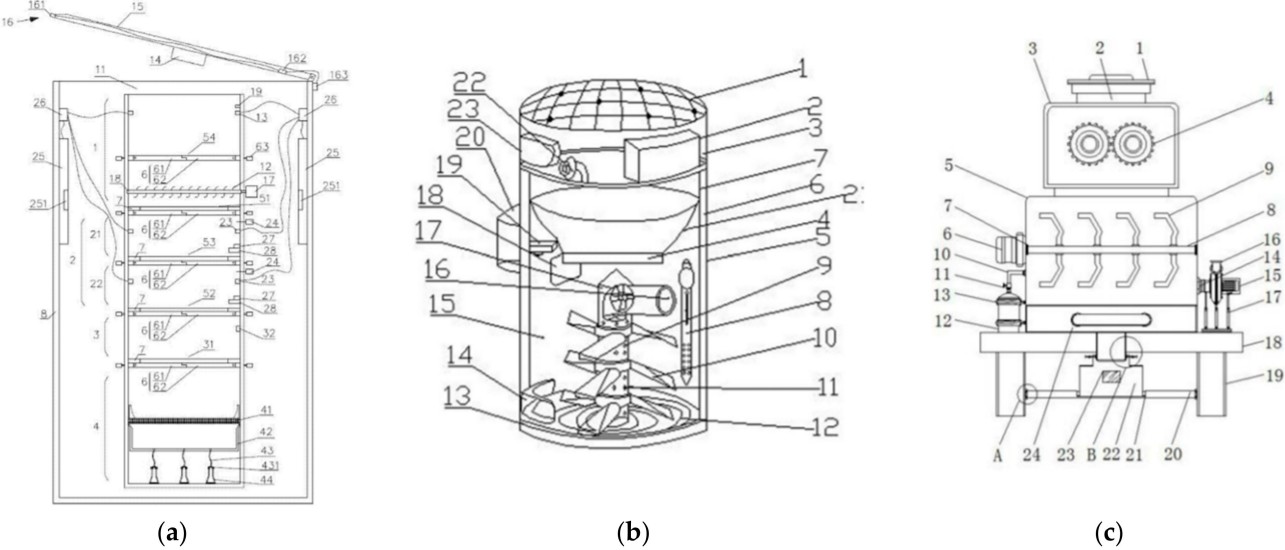

(a)　　　　　　　　　　(b)　　　　　　　　　　(c)

**Figure 21.** (**a**) An aerobic fully automated composter from CN208649153U [101], with details in Appendix A, Table A6; (**b**) an aerobic fully automated composter from CN209555095U [102], with details in Appendix A, Table A7; (**c**) both aerobic and anaerobic fully automated composter from CN212954904U [107], with details in Appendix A, Table A8.

An invention under the anaerobic composting process category, EP2980203A1 [108], describes an anaerobic digester to treat organic waste, consisting of chambers for the decomposition process, components for collecting recoverable products and by-products, a feeder tank for feeding fresh material into the system, a biogas tank, and tanks for collecting treated digestate for use as compost. Each chamber's mixing mechanism consists of a horizontally spinning propeller with two flat paddles. This device controls biogas production and product quality via the monitoring system, which provides the required information throughout the composting process.

A Chinese patent, CN212954904U [107], describes a new type of a fully-automatic household waste composting device with a crushing (no. 4) and deodorizing feature, as

shown in Figure 21c. A pulverizing gear (no. 5) is equipped inside the shredding chamber (no. 3) of the device, powered by a motor (no. 6). Further details on the numberings can be viewed in Appendix A, Table A8. The composting device comes with a deodorizing function and has a strong oxidability; thereby, it is able to decompose the gas emissions and household garbage while also removing the unpleasant odor in the chamber, making it convenient to use. Meanwhile, CN208346058U [109] describes an aerobic-anaerobic composting system for organic waste. Air is supplied into the system through the distributed micro-pores, with a motorized stirring paddle used to rotate the chamber automatically, allowing the efficient mixing of the organic waste while providing sufficient oxygen for the decomposition process. These increase the composting efficiency.

## 7. Conclusions

Waste management is a pressing global issue and, if not treated in a timely and proper manner, can lead to harmful environmental effects. A scientific literature review has been conducted to understand the different methods of managing wastes and their respective benefits and drawbacks. Composting has been found to be a promising technique to mitigate the accumulation of wastes. Despite the large number of scientific studies conducted on composting processes, not a single paper has reviewed the technological advancement of an electrical composter from the patent perspective. Therefore, this paper reviewed and analyzed the technological advancement pertaining to an electrical composter by focusing on filed worldwide patents. A patent landscape review has been conducted by employing a combination of carefully selected keywords and IPC codes to capture the relevant patents, with irrelevant and duplicate patents further removed. The whole review process was carried out in accordance with the PRISMA statement. It has been observed that China is currently leading in publishing electrical composter-related patents, followed by Korea and the United States, especially in the last decade. Analyzing the patent documents has revealed an apparent transition from manual to automated composting technology, with the composting technologies mainly focusing on aerobic composting process and dealing with general organic waste, which can include food waste. The preference towards aerobic composting is due to the smaller capital requirement and the generally faster composting process. The analyses from this patent review paper also indicate the adoption of sensors and relevant control feedback as the technology shifts from conventional manual composting technology to automated composting technology. Insights into technologies designed for different waste types indicate that a small-sized composting system is equally popular compared to a large-sized system, in contrast to other waste types. This indicates that at least for dealing with food/kitchen waste, there is an increased prospect of solving the waste management issues at the source via domestic home-composting systems. Shredding, aeration, and heating are important in the early stages of a composting process. With improper control and monitoring of these parameters, it may result in compost immaturity, which adds extra time before turning into a mature compost. It is noted that there is less work on speeding up the maturity process, which poses an opportunity for the improvement of future composting technology. Undoubtedly, the wide adoption of a domestic electrical composter would also help mitigate the waste management issues associated with organic waste, which is the most common and the highest amount of waste type generated in most countries.

**Author Contributions:** F.A.A. and M.R. did the patent review, patent analysis, reviewing and editing of the manuscript; H.S. and P.E.A. did the structure conceptualization, reviewing and editing of the manuscript. All authors have read and agreed to the published version of the manuscript.

**Funding:** The APC was funded by UBD/RSCH/1.3/FICBF(b)/2020/012.

**Institutional Review Board Statement:** Not applicable.

**Informed Consent Statement:** Not applicable.

**Conflicts of Interest:** The authors declare no conflict of interest.

## Appendix A

The tables below describe the numberings in the figures of the selected patents that are included in this paper.

**Table A1.** An aerobic manual composter from CA2328680C [81].

| Number in Figure | Description |
| --- | --- |
| 1 | Composting device |
| 2 | Receptacle |
| 3 | Rigid side walls of composting container |
| 4 | Base of composting container |
| 5 | Opening of composting container |
| 6 | Hinged closure of composting container |
| 7 | Internal space of composting container |
| 8 | Wheels |
| 9 | Plate |
| 10 | Central aperture of plate |
| 11 | Side supports of plate |
| 12 | Fluid collection chamber |
| 13 | Aerator |
| 14 | Internal air chamber |
| 15 | Lower end internal air chamber |
| 16 | Upper end internal air chamber |
| 17 | Multitude of apertures of aerator |
| 18 | Air supply tube |
| 19 | Air inlet on side wall |
| 20 | Aperture of air supply |
| 21 | Fluid outlet |
| 22 | Clear plastic tube |
| 37 | A lining bag for draining apertures |

**Table A2.** An aerobic manual composter from CA2671248C [82].

| Number in Figure | Description |
| --- | --- |
| 10 | Composter |
| 12 | Body of composter |
| 14 | Frame of the body |
| 16 | First support structure of frame |
| 18 | Second support structure of frame |
| 20 | Connector of frames |
| 22 | Brace of two frames |
| 30 | First side of body |
| 32 | Second side of body |
| 34 | Third side of body |
| 36 | Inwardly curved portion of body |
| 38 | First outwardly curved portion of body |
| 40 | First inwardly extending portions of body |
| 42 | Second outwardly curved portion of body |
| 44 | Distance for separation |
| 46 | Front portion of body |
| 50 | Center section of front portion |
| 54 | Outwardly extending portion |
| 56 | Second inwardly extending portions of body |
| 58 | Lid |
| 62 | Hinges connected to body |
| 64 | Latches secured onto lid |
| 78 | Fasteners connected to front portion |

**Table A3.** An anaerobic manual composter from CN1206189C [90].

| Number in Figure | Description |
| --- | --- |
| 1 | Water pump |
| 2 | Liquid collecting area |
| 3 | Filtering bed fermentation chamber |
| 4 | Outlet of fermenting tank |
| 5 | Motor |
| 6 | Air outlet |
| 7 | Water inlet |
| 8 | Blower |
| 9 | Gas pipe |
| 10 | Sieve plate |
| 11 | Water outlet |
| 12 | Liquid outlet pipe |
| 13 | Shaft |
| 14 | Stirring blades |
| 15 | Liquid inlet |
| 22 | Water pump |

**Table A4.** An aerobic partially automated composter from CA2436322C [88].

| Number in Figure | Description |
| --- | --- |
| 12 | Flat insulated panels |
| 30 | Longitudinal rigid structural elements |
| 35 | Structural hoops |
| 40 | Rollers |
| 41 | Drive shafts |
| 43 | Electric motor |
| 100 | Composting container |

**Table A5.** An anaerobic partially automated composter from CN112354616A [93].

| Number in Figure | Description |
| --- | --- |
| 1 | Processing box body |
| 2 | Driving gear |
| 3 | Buffer piece |
| 4 | Drying box |
| 5 | Guide plate |
| 6 | Crushing box |
| 8 | Second motor |
| 9 | Crushing teeth |
| 10 | Crushing roller |
| 11 | First sieve plate |
| 12 | Brake cylinder |
| 13 | Granulating box |
| 14 | Granulating plate |
| 15 | Second sieve plate |
| 16 | Transmission plate |
| 17 | Transmission gear |
| 18 | Transmission shaft |
| 19 | Transmission wheel |
| 20 | Fan blade |
| 22 | Mounting groove |

**Table A6.** An aerobic fully automated composter from CN208649153U [101].

| Number in Figure | Description |
| --- | --- |
| 1 | Input chamber |
| 11 | Input port |
| 12 | Blades |
| 13 | Second nozzle |
| 14 | Dryer |
| 15 | Bucket cover |
| 16 | Induction control member |
| 161 | Infrared sensor |
| 162 | Controller |
| 163 | Driving part |
| 17 | Vane motor |
| 18 | Vane shaft |
| 19 | Early warning sensor |
| 2 | Fermentation chamber |
| 21 | Preliminary fermentation sub-chamber |
| 22 | Full fermentation sub-chamber |
| 23 | First nozzle |
| 24 | One-way pressure relief valve |
| 25 | Liquid storage tank |
| 251 | Liquid level sensor |
| 26 | Delivery pump |
| 27 | Thermostat |
| 28 | Heating element |
| 3 | Storage room |
| 31 | Drop port |
| 32 | Fertilizer level sensor |
| 4 | Solid-liquid separation component |
| 41 | Filter screen |
| 42 | Liquid container |
| 43 | Conduit |
| 431 | Nested piece |
| 44 | Collection bottle |
| 51 | First opening |
| 52 | Second opening |
| 53 | Third opening |
| 54 | Fourth opening |
| 6 | Opening and closing parts |
| 61 | Left baffle |
| 62 | Right baffle |
| 7 | Sealing ring |
| 8 | Shell |

**Table A7.** An aerobic fully automated composter from CN209555095U [102].

| Number in Figure | Description |
| --- | --- |
| 1 | Solar power generation device |
| 2 | Battery |
| 3 | Upper end of barrel cover |
| 4 | Double-shaft pulverizer |
| 5 | Stainless steel |
| 6 | Sponge |
| 7 | Plexiglass |
| 8 | Oxygen concentration detector |
| 9 | Cylindrical ventilator |
| 10 | Three-layer inclined paddle stirrer |

**Table A7.** *Cont.*

| Number in Figure | Description |
| --- | --- |
| 11 | Air outlets |
| 12 | Electric heating wires |
| 13 | Insulating and thermally conductive silica gel |
| 14 | Compost outlet |
| 15 | Barrel |
| 16 | Flow fan intake port |
| 17 | Axial fan |
| 18 | Natural ventilation port |
| 19 | Ranging infrared sensor |
| 20 | Fungus chaff barrel |
| 21 | Funnel-shaped baffle |
| 22 | Pyroelectric infrared sensor |
| 23 | LED display screen |

**Table A8.** Both aerobic and anaerobic fully automated composter from CN212954904U [107].

| Number in Figure | Description |
| --- | --- |
| 1 | Sealing cover |
| 2 | Feeding port |
| 3 | Crushing chamber |
| 4 | Crushing gear |
| 5 | Operation chamber |
| 6 | Motor |
| 7 | Rotating shaft |
| 8 | Rotating rod |
| 9 | Stirring blade |
| 10 | Oxygen delivery pipe |
| 11 | Valve |
| 12 | Ozone tank |
| 13 | Fixed rope |
| 14 | Delivery port |
| 15 | Fan |
| 16 | Air inlet |
| 17 | Support rod |
| 18 | Partition |
| 19 | Base |
| 20 | Cross bar |
| 21 | Placement slot |
| 22 | Collection box |
| 23 | Sight glass |
| 24 | Drawer |

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
