# Peer review of "Patent Landscape of Composting Technology: A Review"

_inventions, doi:10.3390/inventions7020038_

Round 1

Reviewer 1 Report

Very interesting study. The great effort of the authors to perform the analysis of so many patents should be appreciated. However, with such an extensive study, the authors did not avoid mistakes, which fortunately do not have a major impact on the substantive value of the work.
Detailed remarks:
Line 126 - The equation seems incomplete furthermore there is no source for both equation 1 and equation 2 (line 147).
Line 596 what was the criterion for deeming patents relevant or irrelevant?
The forced aeration used in electric composters is intended to provide oxygen to the microorganisms processing the biological material and even in some cases (line 527) to regulate the temperature of the compost. Excessive aeration can cause a decrease in the moisture content of the composted material which in turn has an adverse effect on the development of active microorganisms. Is this problem not present in the patents analyzed by the authors? 
In Figures 18, 19, 20, 21 the authors show composters without the numbers described. Describing the systems used there or the principle of operation without descriptions makes it much more difficult to understand the innovation of such a solution and significantly reduces the value of the paper.

Conclusions basically do not bring anything new and are a summary. The study would be of greater value if the authors indicated the trends emerging in the development of new technological and design solutions for composters that can be derived from this study. It would be useful, after analysis, to indicate if there are any unsolved problems to which the researchers designing the next new composters should pay attention. In my opinion, the article has yet untapped potential for conclusions.

Reviewer 2 Report

Composting is known to be a time-consuming process that can take time to fully complete the decomposition process and produce compost. This paper is aimed to investigate the Patent Landscape of Composting Technology.

But please see some suggestions:

Figure 1. what is presented, number or %

Figures 8 & 9. The pie chart should have no fill of the chart area because a part of the orange line can not be seen behinde it.

Sincerely,

Round 2

Reviewer 1 Report

I am satisfied with the corrections, although I still think that the authors in the conclusions still have not used the full scientific potential that the article presents. But I am sure that the readers themselves will find interesting information and draw appropriate conclusions. The article is very good and necessary.